



# DUACS DT-2018: 25 years of reprocessed sea level altimeter products.

Guillaume Taburet[1], Antonio Sanchez-Roman[2], Maxime Ballarotta[1], Marie-Isabelle Pujol[1], Jean-François Legeais[1], Florent Fournier[1], Yannice Faugere[1] and Gerald Dibarboure[3]

[1]Collecte Localisation Satellites, Parc Technologique du Canal, 8-10 rue Hermès, 31520 Ramonville-Saint-Agne, France
[2]Instituto Mediterráneo de Estudios Avanzados, C/Miquel Marquès, 21 – 07190 Esporles – Illes Balears, Spain
[3]Centre National d'Etudes Spatiales, 18 avenue Edouard Belin, 31400 Toulouse, France

*Correspondence to*: Guillaume Taburet (gtaburet@groupcls.com)

**Abstract.** For more than twenty years, the multi-satellite DUACS system has been providing Near Real Time (NRT) and Delayed Time (DT) altimetric products. These data are ranging from along-track to multi-mission maps of Sea Level Anomaly (SLA) and Absolute Dynamic Topography (ADT). A reprocessing of 25 years of data, namely: DUACS DT2018, has been carried out and is available through the Copernicus Marine Environment Monitoring Service (CMEMS) and Copernicus Climate Change Service (C3S) since April 2018.

Several changes have been implemented in the DT2018 processing in order to improve the quality of the products. New altimeter standards and geophysical corrections has been used, refined data selection has been implemented and Optimal Interpolation (OI) parameters have been reviewed for global and regional map generation.

Through this paper, an extensive assessment has been carried out. The error budget associated to the DT2018 products at global and regional scales has been refined and the improvements compared with the previous version quantified (DT2014; Pujol *et al.*, 2016). The DT2018 errors at mesoscales are reduced by nearly 3 to 4 % for global and regional products compared to the DT2014. This reduction is much more important in coastal areas (reduction is up to 10%) where it is directly linked to the altimeter geophysical corrections used in the DT2018 processing. Conclusions are very similar concerning geostrophic currents, where error is reduced by 5% and up to 10% in coastal areas.

## 1 Introduction

The multi-mission processing system of altimeter data so called DUACS (Data Unification and Altimeter Combination System) exists since 1997. Since then, it has produced Near Real Time (NRT), with a delay of a few hours to one day, and Delayed Time (DT), with a delay of a few months, altimetric products for the scientific community. The processing unit has been redesigned and regularly upgraded as the knowledge of altimetry processing has been refined (Le Traon *et al.*, 1998; Ducet *et al.,* 2000; Dibarboure *et al.*, 2011; Pujol *et al.*, 2016). Every few years, a full reprocessing is performed by DUACS including all missions and taking into account recent improvements and recommendations from the altimetry community.



This paper presents the latest DUACS DT reprocessing (hereafter DT2018) and focus on improvements that have been conducted since the last version DT2014 (Pujol *et al.*, 2016). Former reprocessing (including DT2014) have been distributed through Aviso from 2003 to 2017. Since May 2015, the whole processing, the operational production and distribution of along-track (level 3) and gridded (level 4) altimeter sea level products has been taken over by the European Copernicus

Program (http://www.copernicus.eu/). Sentinel-3 L3 products are processed on behalf of EUMETSAT, funded by the European Union. The timeseries of the daily DT2018 products starts January $1^{st}$, 1993 and the temporal extension of the sea level record is regularly updated with a nearly 6-month delay with present day. Multi-mission products are based on all altimetry satellites representing a total of 76 cumulated years and 20 missions as shown in Figure 1. The DT2018 reprocessing is characterized by important changes in terms of altimeter standards and data processing compared to the

DT2014 version. These results, highlighted in section 2, have a significant impact on the quality of the sea level products. Two different types of altimeter sea level products are available in the DT2018 version. The first one is dedicated to the retrieval of mesoscale signals in the context of ocean modeling and analysis of the ocean circulation on a global or regional scale. This requires the most accurate sea level estimation at each time step with the best spatial sampling of the ocean by using all mission available. Such dataset is produced and distributed within the Copernicus Marine Service (CMEMS). The

second is dedicated to the monitoring of the long-term evolution of the sea level for climate applications and the analysis of Ocean/Climate indicators (such as the global and regional MSL evolution). This requires a homogeneous and stable sea level record and a steady number of two altimeters is used. Such dataset is produced and distributed within the Copernicus Climate Service (C3S). More details on the differences between the products distributed by both Copernicus Services can be found in section 2.4.

The paper is organized as follows: the DUACS processing from the altimeter standards to L3 and L4 products is considered in section 2. Section 3 and 4 focuses respectively on the quality of the global and regional products at different spatial (coastal, mesoscales) and time scales (climate scales). Finally, the key results and perspectives are covered in section 5.

## 2 Data processing

### 2.1 Altimeter constellation

Seventy-six cumulated years with twelve different altimeters have been used over the twenty-five years period [1993-2017]. The evolution of the altimeter constellation is shown in Figure 1. The most notable change in the constellation with DT2014 concerns Sentinel-3A availability. Extra six months of data (from June 2016 to December 2016) have been added in the system and reprocessed. For some complementary missions, unprocessed data in DT2014 have been taken into account in the DT2018 version. For the most part, it concerns Hayaing-2A between March 2016 and February 2017.



## 2.2 Altimeter standards

DUACS system takes Level 2P (L2P) altimeter products as input data. These data are disseminated by CNES, CLS and EUMETSAT. They include the geophysical standards that allow the calculation of sea level anomalies (i.e. instrumental, geophysical, environmental corrections, Mean Sea Surface - MSS) and a validity flag that is used to remove spurious

measurements. The DUACS DT2018 global reprocessing was an opportunity to take into account new recommendation and new correction from the altimetry community.

The altimeter standards have been selected to be as consistent and homogeneous as possible between the different missions whatever the use (retrieval of mesoscale signals or climate applications). This selection has been made possible in the frame of the phase II of the ESA Sea Level Climate Change Initiative (SL_cci) project between 2014-2017. Within these activities

a tight altimeter standards selection has been carried out (Quartly et *al.*, 2017; Legeais et *al.*, 2018a). Table 1 presents the altimeter standards that have been used in the DT2018 and the changes that occurred compared with the previous version (in bold format). Major changes from the previous version (DT2014) include the implementation of the new GDR-E orbit standard. Orbit standards from Jason-1, Jason-2, Cryosat-2, AltiKa, Jason-3 and Sentinel-2A were upgraded from a GDR-D to GDR-E. New GDR-E standards are reaching a very good quality (Ollivier et *al.*, 2015; AVISO, 2017b), we can note

among others, the following improvement: the evolutions of gravity field has a positive impact on regional MSL error and important reduction of geographically correlated errors that enable to improve the L2 products.

Various corrections have been updated and among them, the new Mean Sea Surface (MSS) and ocean tide model (FES2014) have led to the greatest improvements of the product's quality. Important improvements have been made in the MSS to improve performance at short wavelengths (Pujol et *al.*, 2018a). The sea level in coastal areas and in the Arctic region is also

better retrieved and globally, a strong reduction of the errors has been carried out. Concerning the ocean tide correction, FES2014 is the last version of the FES (Finite Element Solution) tide model being developed in 2014-2016. This new release shows improved results in Deep Ocean, at high latitudes and in shallow/coastal regions (Carrère et *al*., 2016 and Lyard et *al.*, 2016).

## 2.3 Evolution of the DUACS processing

The DUACS processing includes a first preprocessing step to acquire and homogenize the data from the different altimeter. Then, along-track product (L3) and multi-missions gridded products (L4) can be estimated. Finally, derived products are computed and disseminated to the users. This section does not aim to detail the entire data processing system but rather to expose major changes that occurred in this DT2018 version. For an advanced description of the DUACS processing, readers are advised to consult Pujol et *al.,* 2016.



### 2.3.1 Acquisition and preprocessing

Processing sequence in DUACS can be divided into several steps: acquisition, homogenization, input data quality control, multi-mission cross calibration, along-track SLA generation, multi-mission mapping and final quality control.

The acquisition and homogenization processes consist in retrieving altimeter and ancillary data and applying them with the most recent correction, models and reference recommended by expert (as described in section 2.1 and 2.2). This up-to-date selection is available in Table 1.

The Input Data Quality Control is a process linked with the calibration/validation activities carried out for CNES, ESA and EUMETSAT. It is composed of several editing processes to detect and fix spurious measurements and to ensure a long-term stability of L2P products. The up-to-date editing process is described in annual Cal/Val reports for each mission (AVISO 2017c). Since 2014 and learning from experts' experience, great efforts have been performed to refine this global process and notably to adapt some parts to specific regions: high latitude and coastal areas. At high latitudes the idea is to filter an altimeter parameter which has a straight signature on ice, compared to ocean, and then to flag associated data as ice. But filtering solution is global and potential disturbed data out of ice areas can be badly flagged as ice. The proposed evolution consists in using a mask where the chosen filtering solution provides relevant results (Ollivier et *al.*, 2014). The mask is based on the Sea ice concentration product of the EUMETSAT Ocean and Sea Ice Satellite Application Facility (OSI SAF, www.osi-saf.org) and gives us a maximum estimation of ice extent.

In coastal areas, mainly due to a reduced quality of the mean sea surface (MSS) closer than 20 km to the coast, along-track SLA measurements for geodetic and drifting missions were drastically rejected in the DT2014 (Pujol et *al.*, 2016). In DT2018, with improved quality MSS solution, efforts were done to keep as much as possible valid measurements near the coast. The data selection strategy is based on a median filter applied in a 30km band from the coast (Ollivier et *al.,* 2014). Number of valid data usable in DUACS system is now increased in a substantial proportion, especially for geodetic measurements. Figure 2 presents an example of the gain of measurements for the Cryosat-2 geodetic mission in DT2018 over the Mediterranean Sea. 100% of the measurements of geodetic missions are recovered in the 20km band near the coast (all rejected in DT2014 version).

The cross-calibration step makes sure all data from all satellites provide consistent and accurate information. Even if L2 data have been homogenized, they are not always coherent because of various geographically correlated errors ranging from instrument, processing or orbit standards. The first step ensures mean sea level continuity between altimeter missions by reducing global and regional biases for each transition of reference mission (TP-J1, J1-J2 and J2-J3). Then, and in order to minimize geographically correlated errors, two algorithms using empirical process are used: the Orbit Error Reduction (OER) and the Long Wavelength Error Reduction (LWER). The OER is based on a global crossover minimization performed on mono and multi-missions crossovers (Le Traon and Ogor, 1998). The LWER is based on an optimal interpolation process and aim to remove local bias between neighboring for each satellite (Le Traon et *al.* 1998 and Ducet et *al.*, 2000)



### 2.3.2 Along-Track product generation

The along-track generation for repetitive altimeter mission is based on the use of a mean profile (MP) (Dibarboure et *al.*, 2011 and Pujol et *al.*, 2016). These MPs are necessary to co-locate sea surface heights of the repetitive tracks and to retrieve a precise mean reference for the computation of sea level anomalies. The methodology used for the DT2018 MP

computation is the same as in DT2014. Differences come from the upstream measurements, with new altimeter standards used in DT2018 (described in section 2.2), new data selection (see section 2.3.1) and reviewed temporal period for the different altimeters considered. Table 2 introduces the altimeter missions and time periods used to compute the four different MPs that are available along TopexPoseidon/Jason1/OSTM-Jason2/Jason3, TopexPoseidon Interleaved Phase/Jason1Interleaved/Jason2 Interleaved, ERS-1/ERS-2/Envisat/Saral-AltiKa and Geosat Follow On tracks.

Compared to the previous version of the MP, additional measurements collected between 2012 and 2015 were used. They concern OSTM/Jason-2 and SARAL/AltiKa. Since March 2015, AltiKa has been considered as a drifting mission for Delayed-Time product. Therefore, we do not take into account any measurements to compute MP beyond that date. To limit the error of ionospheric correction over the ERS-1/ERS-2/EN/AL Mean Profile, ERS-2 data collected from January 2000 to October 2002 have not been used to compute the MP. Indeed, during this period, the ionospheric activity was much more

intense than between 1995 to 2000.

New DT2018 MPs are defined as close to the coast as possible as illustrated in Figure 3. This improvement is associated with the use of the new MSS and Tide correction and refined valid data selection (see Section 2.2 and 2.3.1). It has a direct positive impact on the along track generation that will benefit of an extended coastal coverage. Globally the comparison of the difference at mono-mission and multi-mission crossovers provides good results in this new version. Compared to the

DT2014 version, we observe at global scale a decrease of the mean of the difference at crossovers around 0.3cm globally and up to 1cm locally (not shown here).

It should be noted that for the Sentinel-3A mission the estimation of a precise MP was not possible for this reprocessing, due to the short time period (i.e. few months) available to compute it. Consequently, data from the Sentinel-3A mission are interpolated onto Theorical Track and the MSS is removed. Since then, a MP has been evaluated (Dibarboure et *al*., in prep

and Pujol et *al.*, 2018b) and Sentinel-3A dataset will be reprocessed in a future CMEMS version in 2019.

For non-repetitive missions (ERS-1 during its geodetic phase, Cryosat-2, Hayaing-2A, Jason-1 geodetic phase, Jason-2 geodetic phase, Saral-AltiKa geodetic phase), no MP can be estimated. The SLA is then derived along the real altimeter tracks using the gridded MSS.

Last step of the along-track processing consists in noise reduction by law-pass Lanczos filtering, and subsampling. This

process remains unchanged compared to the DT2014 version (Pujol et *al.,* 2016).



### 2.3.3 Gridded product generation: multi-mission mapping

The multi-mission mapping process in DUACS is based on an optimal interpolation (OI) technique derived from LeTraon et *al.*, 1998; Ducet et *al.*, 2000 and LeTraon et *al.*, 2003. This method aims at producing regularly gridded products of Sea Level Anomalies by combining measurements from different altimeters.

The last reprocessing DT2014, have shown great improvement on the SLA signal reconstruction mainly offshore (Pujol et *al.*, 2016). The reprocessing DT2018 focused on what had been less emphasized on the previous reprocessing: coastal scale and mesoscale. Specific parameters in the DT2018 OI processing have been optimized to this effect.

The variability of the spatial and temporal scales of the signal have been updated. A particular attention has been put on coastal areas, where spurious peaks of high variability have been reduced. An optimized selection of the data has been

implemented in DT2018 products. The impact is visible at different scales (mesoscale, coastal and climate scale) and over global and regional products. The observations errors have been refined. Errors induced when using the gridded MSS have been updated with the new one for missions that do not use a precise MP. In addition, the *a-priori* knowledge of the signal variance has been updated based on the 25 years of available observations.

Correlation scales remain unchanged for the global and Black Sea products, compared with the ones used in DT2014. They

have been reviewed for the regional Mediterranean products. While set to a constant value (100 km and 10 days) in the DT2014 version, a specific effort has been made to compute precise covariance and propagation models to the DT2018 regional mapping. Spatial scales now range from 75 km to 200 km and temporal scales are set to 10 days. These changes have actively contributed to the improvement of the mesoscale signals' retrieval in the Mediterranean regional products (see section 4).

For the Black Sea processing, OI parameters are now similar to the global ones except for the correlation scales which are still set to 100km and 10days.

### 2.4 Different products for different applications

Two different types of altimeter sea level gridded products are available in the DT2018 version. The first one, produced and distributed within the Copernicus Marine Service (CMEMS), is dedicated to the mesoscale observation. The other one,

produced and distributed within the Copernicus Climate Service (C3S), is rather dedicated to the monitoring of the long-term evolution of the sea level for climate applications and the analysis of Ocean/Climate indicators (such as the global and regional MSL evolution). Different processing parameters leads to these two products. The first difference is related to the number of altimeters used in the satellite constellation.

The mesoscale observation requires the most accurate sea level estimation at each time step with the best spatial sampling of

the ocean. All available altimeters are thus included in the CMEMS products. At the opposite, the temporal stability of the surface sampling is rather required for long-term evolution observation. A steady number (two) of altimeters are thus used in the C3S products. This correspond to the minimum number of satellites required for the retrieval of mesoscale signals in





delayed time condition (Pascual et *al*., 2006 and Dibarboure et *al*., 2011). Within the production process, the long-term stability and large-scale changes are built upon the records from the reference missions (TOPEX-Poseidon, Jason-1, Jason-2 and Jason-3) used in both CMEMS and C3S products. The additional missions (e.g. up to 5 additional missions in 2017) are homogenized with respect to the reference missions and contribute to improve the sampling of mesoscale processes, provide

the high-latitude coverage and increase the product accuracy. However, the total number of satellites strongly varies during the altimetry era and some biases may appear with the introduction of a new satellite flying on a drifting orbit, which may affect the stability of the global and regional MSL from several millimeters (not shown here). Even if the spatial sampling is reduced with fewer satellites, the risk of introducing such anomaly is thus reduced in the C3S products and the stability is improved. In the CMEMS products, the stability is ensured by the reference missions and the mesoscale errors are reduced

due to the improved ocean surface sampling thanks to the use of all satellites available in the constellation.

As a second difference, the reference used to compute the Sea Level Anomalies is a Mean Sea Surface (MSS) for all missions in the C3S products whereas a mean profile of sea surface heights is used along the theoretical track of the satellites with a repetitive orbit in the CMEMS products. The use of MP increases the local accuracy of the sea level estimation (Pujol et *al*., 2018a and Dibarboure et *al*., in prep) but for the C3S production, non-repetitive mission (Cryosat-2) has been used for

a short period of time. Unfortunately, the combined use of MSS and MP for successive missions in the merged product can be at the origin of centimetric bias for regional products (not shown here). So, the systematic use of the MSS for all missions contributes to ensure the MSL stability in the C3S products and the accuracy of the CMEMS products is increased with the use of the mean profiles for repetitive missions.

Differences between CMEMS and C3S product quality are discussed at climatic scale in section 3.4.

**3 DT2018 Global products quality**

The following chapter focuses on the quality of gridded (L4) products. We analyzed sea surface heights and currents derived products at different scales (open ocean, mesoscale and coastal areas) distinguishing different temporal scales (mesoscale to climatic scales). DT2018 L4 products have been compared with DT2014 during the time period 1993-2017. Except when it is mentioned explicitly, the results presented in this section are valid for all DUACS DT2018 products distributed in both

Copernicus services.

**3.1 Mesoscale signals in Along-Track and gridded products**

The mapping process optimization (section 2.3.3) and the new altimeter corrections (section 2.2) have a direct impact on the physical content observed in the gridded products. To characterize this impact, the difference between DT2014 and DT2018 temporal variability is presented in Figure 4. Additional variance is observed for high variability regions in DT2018 products

and is linked to the new OI parametrization. This represents between 2 to 5% of DT2018 the variance. On coastal areas, an important reduction of the variance of the SLA is observed; this being related to both the tidal correction FES2014 and in a





more limited extent to the new Mean Sea Surface. For the tide correction, Lyard et *al.*,2016 and Carrere et *al.*, 2016 have shown a reduction of SLA variance at crossovers nearshore. Pujol et *al.* (2018a) have underlined that the new gridded MSS shows a reduced degradation of SLA near the coast. These improved standards contribute to important local reduction of the SLA error variance (up to 50% alongshore). At high latitude, the difference of variance is important (100cm² to 200cm²) and

is linked to the new MSS correction. Indeed, Pujol et *al.* (2018a) have shown that CNES_CLS 2015 MSS improves coverage in arctic and the shortest wavelength at high latitude.

Compared to the DT2014, the new version has more intense geostrophic current in western boundary currents. This has a direct impact on the Eddy Kinetic Energy (EKE) derived from these products. Figure 5 presents the spatial difference of the mean EKE over global ocean between DT2018 and DT2014 products and also their temporal evolution. As observed before

in the difference of SLA variance, we clearly see higher energy in high variability areas. This represents a 2% increase EKE in DT2018. However, in the equatorial band (±20°N), the EKE in the DT2018 is less important (-17%). This is linked with the evolution of the noise measurement considered in the mapping process for all satellites. The consistency between altimeter geostrophic current and independent measurements is significantly improved in this area as discussed in section 3.2. On coastal area, the DT2018 version presents less spurious peaks of high EKE. As already stated, this is linked with the

improved altimetric correction and the variance SLA reduction. Considering the mean EKE time series, a global reduction of 26 cm² (17%) is observed for the DT2018 dataset. It is directly linked with the equatorial EKE reduction. What is also important to note is that the temporal evolution of the EKE in these products is less peaky than in DT2014. This illustrates that there is fewer isolated anomalies (mostly coastal) in the new DT2018 products.

The accuracy of the gridded SLA is estimated by comparing SLA with independent along-track measurements. Maps

produced by merging only two altimeters (C3S products) are compared with SLA measured along track from the tracks of another mission kept independent of the mapping process (see Pujol et *al.*, 2016 for full methodology).  TP interleaved is compared with gridded product that merges J1 and EN over the year 2003-2004. It is then important to note that these results are much more representative of "two-sat-merged" gridded products. The "all-sat-merged" products can usually benefit from an improved sampling when three to six altimeters are used. Thus, the errors described here should be considered as the

upper limit. Table 3 summarizes the results of the comparisons over different areas.  The gridded product error for mesoscale wavelengths ranges between 1.4 cm² (for low variability area) and 37.7 cm² (for high variability region). The improvement of DT2018 compared with DT2014 is global: Offshore, the improvement is quite low (3%) and is associated with the enhanced version of the mapping parameter of the OI. In coastal area, the improvements are more significant (10%) and linked with the use of the new Tide correction (FES2014) and, to a lesser extent, with the MSS and MPs.

**3.2 Geostrophic current quality**

DT2018 absolute geostrophic current has been assessed using drifter data for the time period 1993-2017. AOML (Atlantic Oceanographic & Meteorological Laboratory) database has been used for the comparison (Lumpkin et *al.* 2013). These *in-situ* data are corrected from Ekman drift (Rio et *al.,* 2011) but also from wind if drifters' drogue has been lost (Rio et *al.*,



2012) so as to be compared with altimetry absolute geostrophic current. Positions and velocities of drifters are interpolated using a 3-day low-pass filter in order to remove high-frequency motions. Absolute geostrophic current derived from altimetry products are then interpolated onto drifter positions for comparison.

As the previous version (Pujol *et al.*, 2016), the comparison reveals that DT2018 altimetry products underestimate absolute geostrophic current. Figure 6 shows the RMS difference between DT2018 geostrophic current and drifters. Mean RMS is nearly 10 cm/s and main errors are located nearshore and in high variability region with peaks higher than 20 cm/s. Taylor skill scores (Taylor, 2001) have been computed for the zonal and meridional components of the current in DT2018. This assessment enables to consider both variance and RMS of the signal. Results are quite strong: 0.89 for the zonal and 0.87 for the meridional component.

The Table 4 summarizes mean rms of the differences between altimeter maps and drifter measurements over different areas for the DT2018 and DT2014 version. DT2018 products are more consistent with drifter measurements than the DT2014 version. The improvement is clearly visible in the intra-tropical band. The Variance of the differences with drifters is reduced by 7% in this area. Additional noise-like signal, previously introduced in the DT2014 version and leading to a degradation of the consistency with drifter measurement (Pujol et *al*., 2014) is now corrected in the DT2018 version. This is directly linked to the change of the mapping parameters used for this version (see section 2.3.3). Significant improvement is also observed in coastal areas with a reduction of the variance of the differences with drifter measurement reaching nearly 15% (Table 4). Elsewhere, the variance difference reduction ranges between 4 and 7%.

## 3.3 Coastal areas

As described in sections 2.3.1 and 2.3.2 the new DUACS DT2018 processing has an important impact on coastal areas. The clearest impact is the major gain of points from every non-repetitive missions and missions not having a MP. We gain all points no further than 20 km of the coast for these six missions over 16 years in total. There is also an improvement for repetitive missions since in average we gain points nearshore (Figure 3). Overall, all missions have more measurements available in DT2018 compared to the previous version.

Specific efforts were done in the DT2018 processing to improve the products quality near the coast. Choice of up-to-date standards, specifically ocean-tide and MSS (see section 2.2), clearly contribute to the quality of the altimeter measurement near the coast. Additionally, refined data selection (see section 2.3.1) significantly increase the data availability in the in the band 20km close to the coast. Finally, review of the mapping parameters (section 2.3.3) also contribute to the improved quality of the gridded products in the coastal area.

Previous comparisons between gridded maps and independent measurement underlined the positive impact of the DT2018 processing in the coastal area. Compared with results obtained with DT2014 version, we observe with DT2018 a reduction of the variance of the differences between gridded SLA products and independent along-track measurements by nearly 10% (Table 3, Section 3.1), and a reduction of the RMS of the differences between altimeter geostrophic current and drifter measurement by nearly 15% (Table 4, section 3.2).





The assessment of the gridded products in coastal areas was completed with comparison with tide-gauges (TG) measurements. We have used monthly mean TG measurements from the PSMSL network (Permanent Service for Mean Sea Level, PSMSL, 2016) from 1993 to 2017. We considered only long-term monitoring stations with a lifetime greater than 2 years. Sea surface Height measured by TG is compared with gridded SLA by considering the maximum correlation with the nearest neighboring pixel (Valladeau *et al.*, 2012 and AVISO, 2017a). In Figure 7, the variance of the difference between DT2018 altimetric products and TG measurements is compared with that obtained from the differences using DT2014 altimetric products. The results show a global reduction of the variance when DT2018 are used. There is a clear improvement along the Indian coast, Oceania and northern Europe. A local degradation can be observed along the coast of Spain and along the Western coast of United States. These degradations that are not observed in other diagnoses like independent along track measurement are not understood yet.

**3.4 Climate scales**

The global mean sea level (MSL) is a key indicator of climate change and can be computed from the time series of the box-averaged along-track measurements of the reference missions only (Ablain *et al.*, 2017). The global MSL can also be derived from the DUACS L4 merged gridded sea level products distributed by both marine and climate Copernicus services (e. g. Figure 8, left). Considering the same products version and period of computation, these three estimates of the global MSL are considered to be equivalent since almost the same altimeter standards are used to compute the sea level anomalies and for all products, the long-term stability is ensured by the same reference missions. The remaining observed global MSL differences (~0.17mm/year) are not significant given the uncertainty considered on different scales (uncertainty in the GMSL trend is approximately of 0.5 mm/yr. at the 90% confidence level given Legeais et *al.*, 2018b). Note that as aforementioned (section 2.4), the situation is not the same on a regional scale where differences can be found according to the product used (CMEMS/C3S) for the MSL computation.

When computing area-averaged MSL time series, users are advised that the DUACS products are not corrected for the effect of the Glacial Isostatic Adjustment (GIA) due to the post glacial rebound and a GIA model should be used to estimate the associated sea level trends.

In addition, between 1993 and 1998, the global MSL has been known to be affected by an instrumental drift in the TOPEX-A measurements which has been quantified by several studies (Watson et *al.*, 2015; Beckley et *al.*, 2017; Dieng et *al.*, 2017). The altimeter sea level community agrees that it is necessary to correct the TOPEX-A record for the instrumental drift to improve the accuracy and the uncertainty of the total sea level record. However, there is not yet consensus on the best approach to estimate the drift correction at global and regional scales. The DUACS altimeter sea level products are not corrected for the TOPEX-A drift, waiting for the on-going TOPEX reprocessing by CNES and NASA/JPL but the users can apply their own correction. Adjusting for this TOPEX-A anomaly create a GMSL acceleration of 0.10mmyr$^{-2}$ for the 1993–2017 time span that does not appear otherwise (WCRP 2018).



Figure 8 (left) shows the global Mean Sea Level temporal evolution and associated trend computed with the new DT2018 and former DT2014 versions of the DUACS C3S products. With the latest version, the global mean sea level trend is of 3.3 mm/yr. (including a GIA correction of -0.3 mm/yr.) and the origin of the associated uncertainty is discussed by Legeais et *al.* (2018b). The map of the differences of the local MSL trend derived from the latest and previous products versions (Figure 8, right) displays a pattern dominantly associated with the difference of orbit standards used in both versions of the products (GDR-E versus GDR-D, see Table 1). Such result is confirmed by the comparison of the altimeter products with the independent measurements of dynamic heights derived from in-situ Argo profiles (Valladeau et *al.,* 2012; Legeais et *al.*, 2016).

## 4 DT2018 Regional product quality

### 4.1 SLA field quality

As previously discussed for the Global ocean products, the quality of the regional gridded SLA products is estimated by comparison with independent altimeter along-track and tide gauge measurements.

The Figure 9 shows the spatial distribution of the RMS of the differences between regional DT2018 SLA gridded products and independent along-track (Topex/Poseidon Interleaved along track measurements over the period [2003-2004]). Main statistics on these comparisons, as well as comparison with previous DT2014 version, are given in Table 5. Contrary to the processing applied for Global product assessment, the short length of the main part of the tracks available over these Seas does not allow us to accurately filter the signal in order to focus specifically on mesoscale signals. The results obtained show that For the Mediterranean product, the main errors are located on coastal areas and in the Adriatic and Aegean Seas with RMS values ranging from 6 to 9 cm. The Black Sea products present also higher errors on coastal areas. The mean rms of the differences between gridded products and along-track measurement reaches nearly 17cm² (23 cm²) over the Med. (Black) Sea. This value is higher than the mean error observed over low variability area in the Global ocean (Table 3), mainly due to the different wavelengths addressed in these comparisons. Compared to the previous regional version DT2014, the error is reduced by 4.2% (3.5%) for the Med (Black) Sea. It is important to note that these results are representative of the quality of the gridded products when only two altimeters are available. These products can be considered as degraded products for mesoscale mapping since they use minimal altimeter sampling.

Consistency with monthly Tide Gauges measurements (Figure 10) is improved with the regional DT2018 Mediterranean gridded product from the Balearic to Ligurian Seas as well as in the Adriatic Sea compared to the previous version. In some other coastal areas, degradation is however observed especially in the Aegean Sea and along the Sicilian coast.

### 4.2 Geostrophic current quality in the Mediterranean Sea

DT2018 regional absolute geostrophic current in the Mediterranean basin has been assessed using drifter data for the period [1993-2017]. Drifters released in the Mediterranean Sea in the frame of AlborEx (Pascual et *al.*, 2017) and MEDESS-GIB





(EU MED Program; http://www.medess4ms.eu/ ; Sotillo et *al.,* 2016) multi-platform experiments as well as other experiments gathered in the In Situ Thematic Centre (INS TAC) products from CMEMS were used. The processing of these data is analogue to that applied to the global product (see section 3.2).

Table 6 summarizes the main statistical results for the whole basin. The DT2018 regional product presents a correlation coefficient with drifter data 4% larger than that obtained when using the DT2014 regional product. Moreover, the errors in the former are slightly reduce by 1% whilst its improvement in the explained variance reaches 14%.

We repeated the analysis but for the different dynamical sub-regions of the basin (see Figure 11.a) reported by Manca et *al.* (2004). This differentiation is based on the typical permanent features in the upper 200 m of the water column. Overall, comparisons between geostrophic velocities derived from the DT2018 regional product and absolute surface velocities retrieved by the drifters (Figure 11. b – e) present a correlation coefficient larger than 0.40 in most of the boxes. Correlations larger than 0.50 are mainly located in the southernmost part of the basin where a stronger mesoscale activity occurs; namely the Alboran Sea (DS1), the Algerian Basin (DS3 and DS4), the Sardinian Channel (DI1), the Sicily Strait (DI3), the Ionian Sea (boxes DJ7, DJ8 and DJ5), and the Cretan passage (DH3). The overall RMS difference between both datasets ranges between 8 – 11 cm/s whilst it reaches 20 cm/s in DS1 due to the strong dynamics of this area. Slightly large errors are obtained when comparing the DT2014 product with drifter observations (figure not shown). Furthermore, drifter data collected in boxes DS1, DS3 and DS4 present the largest variability due to the aforementioned mesoscale activity. This fact is also reflected in the two altimetry products, which present there the largest variance values in the basin.

Overall, the correlation coefficient between DT2018 regional product and in-situ drifter data improves between 5-10% with respect to that obtained when using the DT2014 product (Figure 11.g). Here, positive values denote an improvement of DT2018 over DT2014. This fact is mainly observed in areas of strong mesoscale activity. Moreover, the errors (Figure 11.f) reduce around 2% in the northernmost part of the western basin and Adriatic Sea. However, negative values lower than 2% (slightly lager errors when using DT2018) are observed in the Algerian Basin and most of the eastern basin. The main improvement of DT2018 with respect to DT2014 lies on the variance explained (Figure 11.h), which presents values nearly 20% (10%) larger in the former in some places of the western (eastern) basin. This is due to a better capture of the mesoscale activity. This improvement is not observed in the northernmost part of the basin, where a lower mesoscale activity occurs.

## 5 Discussions and Conclusions

More than 25 years of Level-3 and Level-4 altimeter products have been reprocessed and delivered as the DT2018 version. This reprocessing takes into account the most up-to-date altimeter correction and also include changes in the parameters involved in the mapping processing. These changes impact the SLA signals at multiple temporal and spatial scales.

An important change concern the gridded altimeter sea level product that are available in the DT2018 version. They are produced and distributed through two different Copernicus Service that correspond to different applications. Through CMEMS, maps that included all altimeter missions available are distributed. They give the most accurate sea level



estimation with the best spatial and temporal sampling of the ocean at all time. Through C3S, maps that include only two satellites are used to compute the most homogeneous and stable sea level record though time and space. Sea Level C3S products are dedicated to the monitoring of the long-term evolution of the sea level for climate applications and the analysis of Ocean/Climate indicators (such as the global and regional MSL evolution).

Other changes have been implemented in the DT2018 processing; the altimeter standard and geophysical corrections are up-to-date with expert recommendation, mapping parameters have been refined including spatial and temporal correlation scale and measurements errors. We also focused on the improvement of coastal editing to gain many relevant sea level data, mainly for drifting altimeter. Additional sea level data have been used compared to the DT2014, especially Sentinel-3A measurements that have been used over a 6-month extended period.

Having discussed these important changes, we have focused on the description of the impact on gridded sea level products. The SLA variability has been increased in energetic area (from 5 to 10%) and reduced locally along the coast (up to 50%). A 10% EKE decrease in the equatorial band is also observed and linked to the refined measurements errors in the area.

To realize independent comparisons, we have used unrelated *in-situ* measurements. Geostrophic currents have been examined and are still underestimated compared to the in-situ observations. Nevertheless, compared to the DT2014 version,

offshore improvements (+4-5%) and coastal improvement (+10%) have been shown using independent drifters' data. Independent along-track sea level comparison and Tide Gauges comparisons have strengthened these conclusions.

Regional products are also improved with DT2018, taking advantage of the altimeter new standards and processing. The SLA gridded product errors in the regional products are decreasing from 3% to 4% when estimated using independent along-track measurements.

Limitations exposed by Pujol et *al.* (2016) are still valid and the errors observed in the retrieval of mesoscale features also highlight the L4 product spatial resolution capability. To estimate the spatial resolution of the gridded products, an evaluation has been carried out based on a spectral coherence approach. A full description of this approach can be found in Ballarotta et *al.*, (in prep.).

Many products are derived from these global and regional gridded products and are strongly affected  by the their quality:

Lagrangian product (FSLE d'Ovidio et *al.* 2015), or eddy tracking (https://www.aviso.altimetry.fr/en/data/products/value-added-products/global-mesoscale-eddy-trajectory-product.html) are a prominent examples.

Medium term developments concern new Level-3 products that will be dedicated to data assimilation and CMEMS Monitoring Forecasting Centre. These products will be new in Delayed-Time mode. The Mean Dynamic Topography will also be updated, and the Black Sea area will be integrated. Finally, a new European regional product will substitute to the

current Mediterranean and Black Sea products.

In the coming years, DUACS will face important challenges with the arrival of new altimeter mission. SWOT for example will observe fine-scale dynamics, with swath SSH observations (Morrow et al., 2018), that need to be integrated in the DUACS system. To do so, the next step will consist in moving toward higher resolution for along track and gridded product.



New mapping technique should also be taken into consideration and are currently studied as dynamical advection (Rogé *al.*, 2017, Ubelmann et *al.*, 2016).

## 6 Data availability

The datasets are available from the CMEMS web-portal (http://marine.copernicus.eu/services-portfolio/access-to-products/)

5  and the C3S data store (https://cds.climate.copernicus.eu). Level 2 (GDR) input data are provided by CNES, ESA, EUMETSAT and NASA.

The MEDESS-GIB dataset is available through the PANGAEA (Data Publisher for Earth and En- vironmental Science) repository, with the following doi:10.1594/PANGAEA.853701. The AlborEx dataset is available at the SOCIB web page (http://www.socib.eu).

## 7 Acknowledgements

The DT2018 reprocessing exercise has been supported by CNES/SALP project, CMEMS and C3S services funded by the European Union. Global L3 Sentinel-3 production is coordinated by EUMETSAT and funded by the European Union.



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



**Table 1: Altimeter standards used in DT2018. Standard changes with the DT2014 solution are underlined in bold format.**

| | J3 | J2 | J1 | TP | ERS-1 | ERS-2 | EN | GFO | C2 | AL | H2A | S3A |
|---|---|---|---|---|---|---|---|---|---|---|---|---|
| Orbit | GDR-E | **GDR-E** | | **GFSC STD15 until cycle 365, STD12 afterwards** | Reaper [Rudenko et al., 2012] | | GDR-D | GSFC | **GDR-E** | | **GDR-D** | GDR-E |
| Sea State Bias | Non-Parametric SSB [Tran et al., 2012] | **SSB issued from GDR-E** | Non-parametric SSB [Tran et al., 2010] | BM3 [Gaspar et al., 1994] | Non-parametric [Mertz et al., 2005] | **Non-Parametric SSB [Tran et al., 2012]** | Non-Parametric SSB [Tran et al., 2010] | Non-Parametric SSB from J1 with unbiased sigma0. | **Non-Parametric SSB [Tran et al., 2012]** | **Non-Parametric SSB from J1** | Non-Parametric SSB [Tran et al., 2012] | |
| Ionospheric | Filtered dual-frequency altimeter range measurements [Guibbaud et al., 2015] | **Filtered dual-frequency altimeter range measurements [Guibbaud et al., 2015]** | Filtered dual-frequency altimeter range measurements [Guibbaud et al., 2015] | Reaper NIC09 model [Scharroo et al., 2010] | **Cycle<37 Reaper NIC09 model [Scharroo et al., 2010] Cycle>36 GIM [Ijima et al., 1999]** | Dual-frequency altimeter range measurement [Guibbaud et al., 2015] (6≤cycles≤64)/GIM [Ijima et al., 1999] Corrected for 8mm bias (c≥65) | GIM [Ijima et al., 1999] | | **GIM [Ijima et al., 1999]** | | Filtered dual-frequency altimeter range measurements [Guibbaud et al., 2015] | |
| Wet troposphere | From J3-AMR radiometer | **Neural Network correction [Keihm et al. 1995]** | **JMR issued from GDR-E** | **GNSS derived Path Delay [Fernandes et al., 2015]** | | **Neural Network correction (5 entries) [Obligis et al., 2009 and Picard et al., 2015]** | From GFO radiometer | From ECMWF model | **Neural Network correction (5 entries) [Obligis et al., 2009 and Picard et al., 2015]** | **From ECMWF model** | From S3A-AMR radiometer | |
| Dry troposphere | Model based on ECMWF Gaussian grids | Model based on ECMWF rectangular grids | Model based on ERA-INTERIM | | | Model based on ECMWF Gaussian grids | Model based on ECMWF rectangular grids | Model based on ECMWF Gaussian grids | **Model based on ECMWF Gaussian grids** | Model based on ECMWF Gaussian grids | | |
| DAC | MOG2D High frequencies forced with analysed ECMWF pressure and wind field [Carrere et al., 2003; operational version used, current version is 3.2.0] + inverse barometer Low frequencies | | MOG2D High frequencies forced with analysed ERA-INTERIM pressure and wind field + inverse barometer Low frequencies | | | MOG2D High frequencies forced with analysed ECMWF pressure and wind field [Carrere et al., 2003; operational version used, current version is 3.2.0] + inverse barometer Low frequencies | | | **MOG2D High frequencies forced with analysed ECMWF pressure and wind field [Carrere et al., 2003; operational version used, current version is 3.2.0] + inverse barometer Low frequencies** | MOG2D High frequencies forced with analysed ECMWF pressure and wind field [Carrere et al., 2003; operational | |



| | version used, current version is 3.2.0] + inverse barometer Low frequencies |
|---|---|
| Ocean tide | **FES2014 [Carrere et al., 2015]** |
| Pole tide | **[Desai et al., 2015]** |
| Solid earth tide | Elastic response to tidal potential [Cartwright and Tayler, 1971], [Cartwright and Edden, 1973] |
| Mean Sea Surface | **CNES-CLS-2015 [Pujol et al., 2018a]** |



**Table 2: Time periods and Cycles used to compute Mean Profile in the DT2018 version.**

| | Satellite used in Mean Profile computation | Periods used in Mean Profile computation | Cycles |
|---|---|---|---|
| **Topex/Poseidon – Jason-1 – Jason-2 – Jason-3** | Topex/Poseidon | January 1993 – April 2002 (9 years) | 11 – 353 |
| | Jason-1 | April 2002 – October 2008 (6 years) | 10 – 249 |
| | OSTM/Jason-2 | October 2008 – December 2015 (7 years) | 10 – 273 |
| **Ers-1 – Ers-2 – Envisat - AltiKa** | Ers-2 | Mai 1995 – January 2000 (5 years) | 1 – 49 |
| | Envisat | October 2002 – October 2010 (8 years) | 10 – 94 |
| | AltiKa | March 2013 – March 2015 (2 years) | 1 – 22 |
| **Topex/Poseidon Interleaved orbit – Jason-1 Interlevead orbit – Jason-2 Interlevead orbit** | Topex/Poseidon Interleaved orbit | September 2002 – October 2005 (3 years) | 368 – 481 |
| | Jason-1 Interlevead orbit | February 2009 – March 2012 (3 years) | 262 – 374 |
| **Geaosat Follow On** | Geaosat Follow On | January 2000 – September 2008 (8 years) | 37 – 222 |

**Table 3: Variance of the differences between gridded (L4) DT2018 two-sat-merged products and independent TP interleaved along-track measurements for different geographic selections (unit = cm²). In parenthesis: variance reduction (in %) compared with the results obtained with the DT2014 products. Statistics are presented for wavelengths ranging between 65-500 km and after latitude selection (|LAT|<60°).**

| | TP [2003-2004] |
|---|---|
| **Reference area\*** | 1.4 (-0.3%) |
| **Low variability (<200 cm²) & offshore (distance coast >200 km) areas** | 5.0 (-3.0%) |
| **High variability (>200 cm²) & offshore (distance coast >200 km) areas** | 37.7 (-3.1%) |
| **Coastal areas (distance coast < 200km)** | 8.2 (-10.1%) |

*\*The reference area is defined by [330, 360°E]; [-22, -8°N] and corresponds to a very low-variability area in the South Atlantic subtropical gyre where the observed errors are small*



**Table 4: Variance of the differences between gridded geostrophic current (L4) DT2018 products and independent drifter measurements (unit = cm2/s2). In parenthesis: variance reduction (in %) compared with the results obtained with the DT2014 products. Statistics are presented for latitude selection (5°N<|LAT| < 60°N).**

|  | Zonal | Meridional |
|---|---|---|
| **Reference area*** | 44.3 (-1.8%) | 33.4 (-0.9%) |
| **Dist coast > 200km & variance < 200 cm²** | 91.6 (-6.1%) | 88.6 (-6.7%) |
| **Dist coast > 200km & variance > 200 cm²** | 229.6 (-4.3%) | 260.5 (-4.5%) |
| **Dist coast < 200km** | 189.7 (-14.7%) | 195.3 (-15.5%) |

*\*The reference area is defined by [330, 360°E]; [-22, -8°N] and corresponds to a very low-variability area in the South Atlantic subtropical gyre where the observed errors are small*

**Table 5: Variance of the differences between gridded (L4) DT2018 two-sat-merged regional Mediterranean (first line) and Black sea (second line) products and independent TP interleaved along-track measurements without filtering over the time period 2003-2004 (unit = cm²). In parenthesis: variance reduction (in %) compared with the results obtained with the DT2014 products.**

|  | TP [2003-2004] unfiltered |
|---|---|
| **Mediterranean Sea product** | 16.7 (-4.2%) |
| **Black Sea product** | 23.2 (-3.5%) |

**Table 6: RMSE (m/s) and correlation coefficient between the absolute geostrophic velocities derived from DT-2018 regional products for the Mediterranean Sea; and absolute surface velocities as obtained from drifters collected in the basin. The variance of the datasets (m²/s²) and the data used to conduct the comparison are also displayed.**

|  | DT-2018 regional | DUACS-DT2018 improvements |
|---|---|---|
| **R** | 0.49 | 4 % |
| **RMS diff (m/s)** | 0.12 | 1 % |
| **variance drifter (m²/s²)** | 0.017 | - |
| **variance altimetry (m²/s²)** | 0.008 | 14 % |





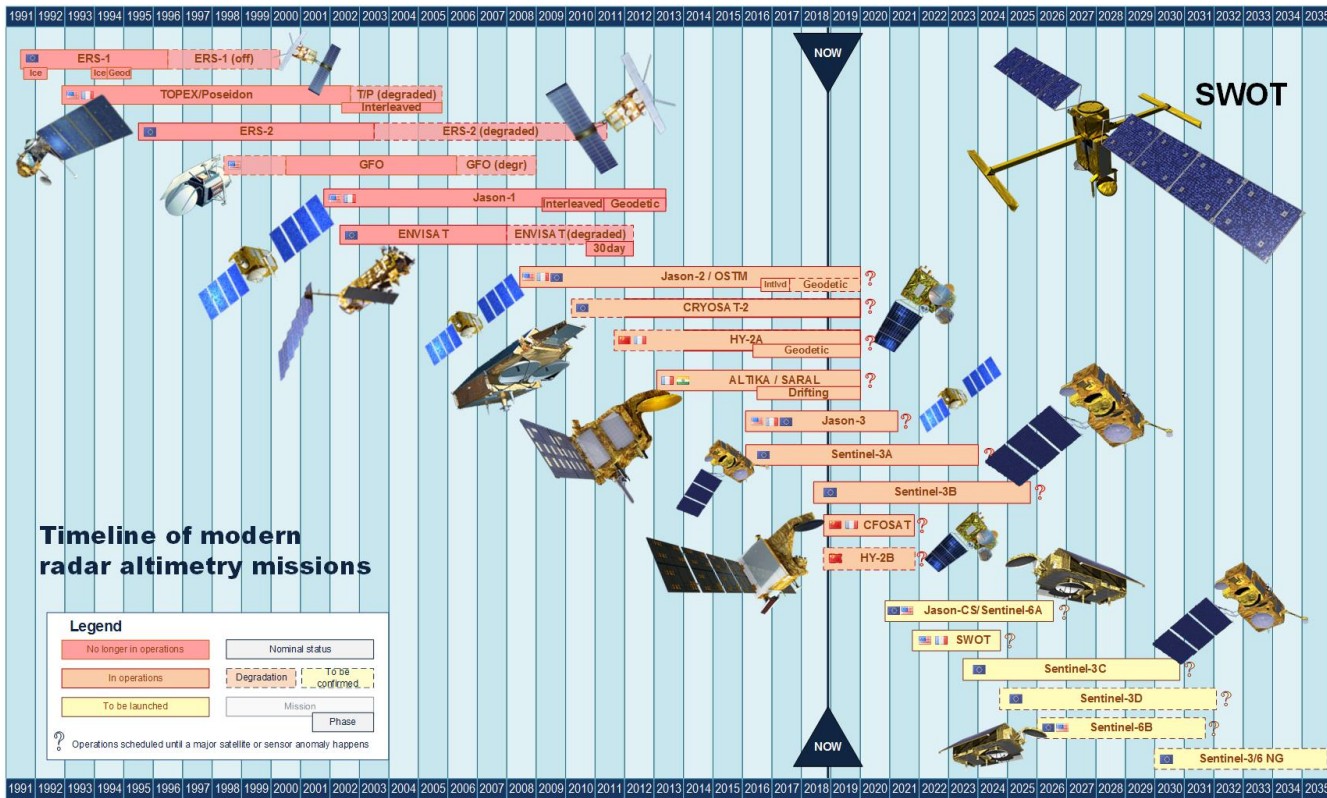

**Figure 1: Timeline of the altimeter missions used in the multi-mission DUACS-DT 2018 system.**



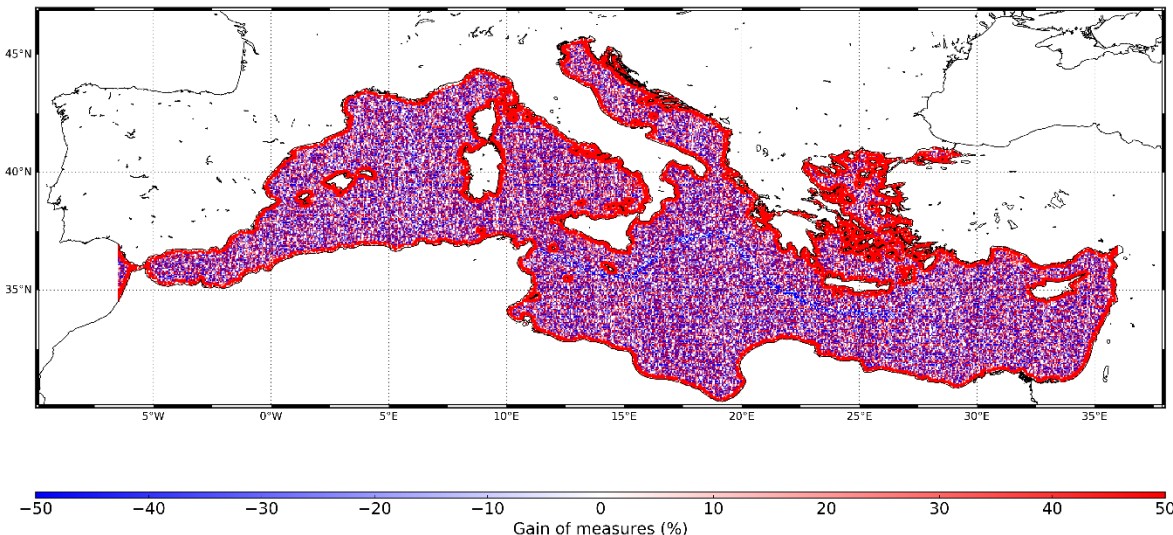

**Figure 2: Gain in percent of Cryosat-2 L2P data in DT-2018 version compared to the DT-2014 version for the Mediterranean Sea product. Gain of points with the DT-2018 version are in red, Loss in blue.**

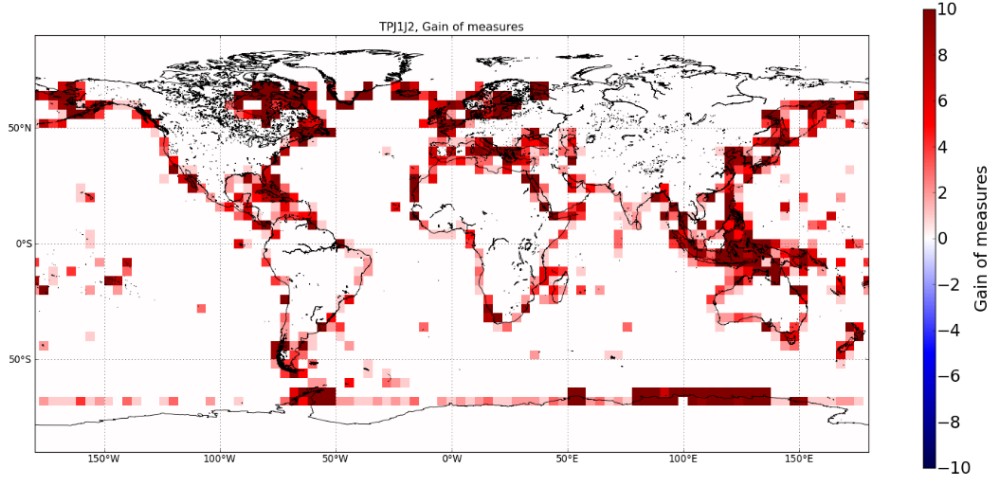

**Figure 3: Gain of measurements in the Topex/Poseidon-Jason1-OSTM/Jason-2 Mean Profiles used in DT-2018 versions compared to the DT-2014. Gain of points in the DT-2018 version are in red, Loss ones in blue.**



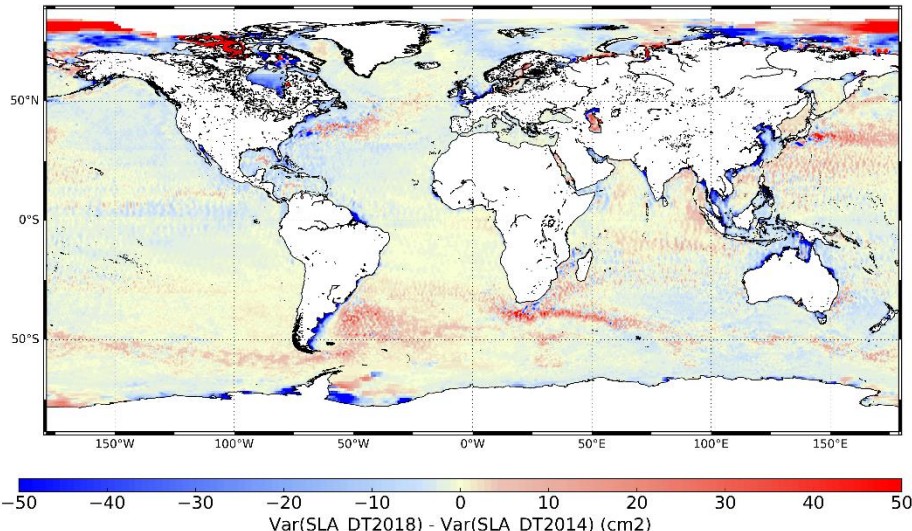

**Figure 4: Difference between SLA variance observed with DT2018 and DT2014 gridded products over the time period 1993-2017. Units: cm².**

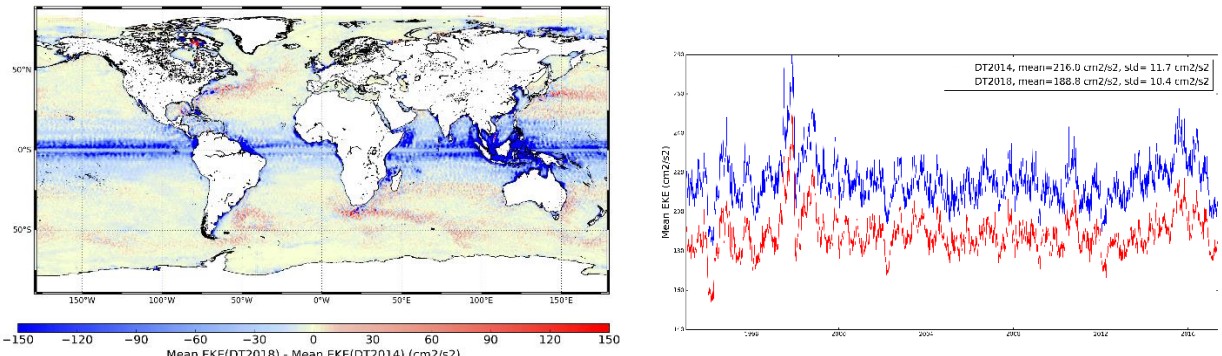

**Figure 5: EKE mean difference between DT2018 and DT2014 gridded products (left frame) and EKE time series (right frame)**
10 **computed from DT2014 (blue line) and from DT2018 (red line) SLA over the time period 1993-2017. Units: cm²/s².**



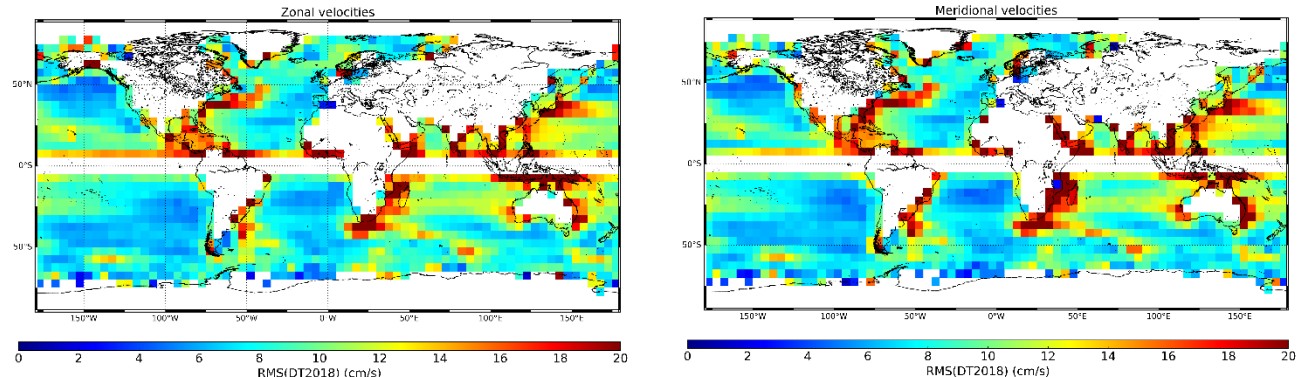

**Figure 6: Zonal (left) and meridional (right) RMS of the difference between DUACS DT2018 absolute geostrophic current and drifter's measurements over the period [1993-2017]. Statistics have been computed in boxes of 5°x5°. (units: cm/s). Difference of the variance of the altimeter SLA minus Drifters SLA differences, using successively DT2018 and DT2014 SLA gridded products.**
5  **Negative values mean that the SLA differences between altimetry and drifters are reduced when considering DT2018 products.**

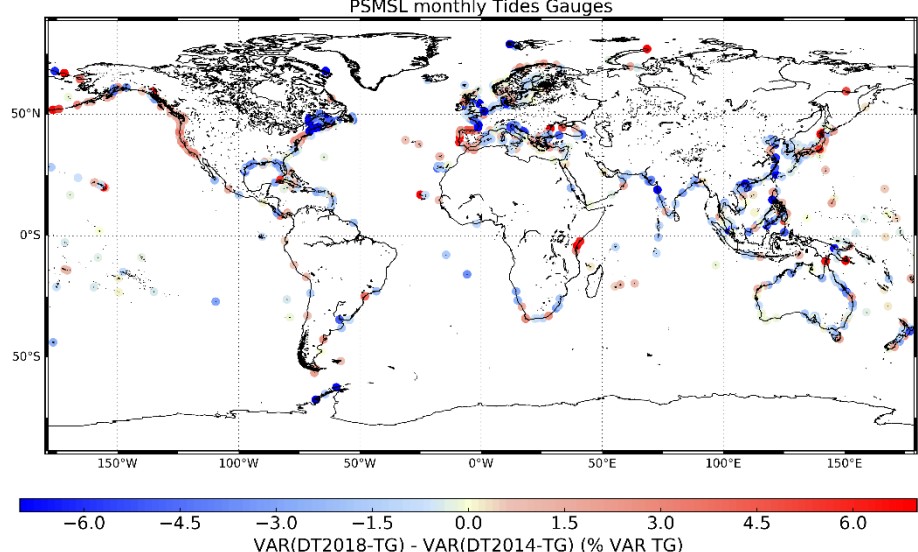

**Figure 7: Difference of the variance of the altimeter SLA minus TG SLA differences, using successively DT2018 and DT2014 SLA gridded products. Monthly Tide Gauges come from PSMSL network. Negative values mean that the SLA differences between altimetry and TGs are reduced when considering DT2018 products.**





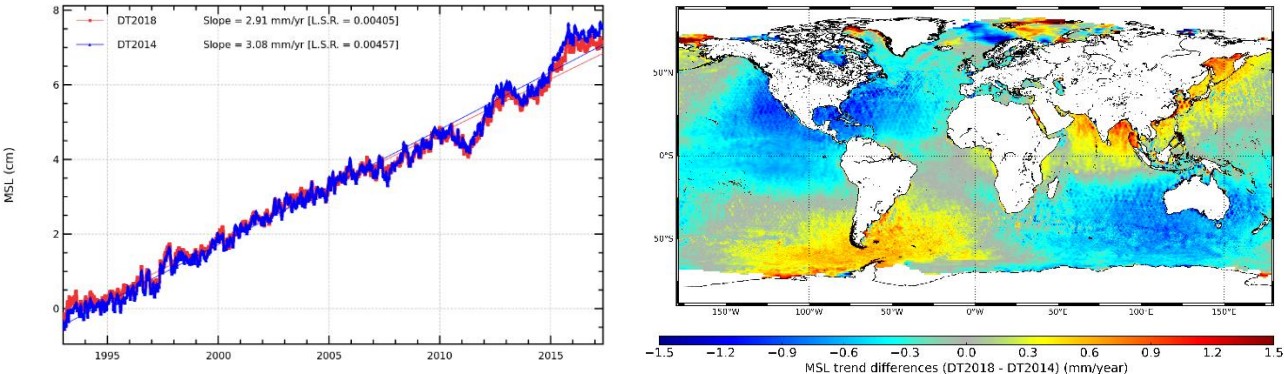

**Figure 8: Left panel: temporal evolution of the global MSL estimated from DT2018 (red line) and DT2014 (blue line) gridded SLA products. The annual and semi-annual signals have been adjusted and no GIA correction has been applied. Right panel: map of the differences of the local MSL trend estimated from the DT2018 and DT2014 gridded SLA products. MSL estimated over the 1993-2017 period.**

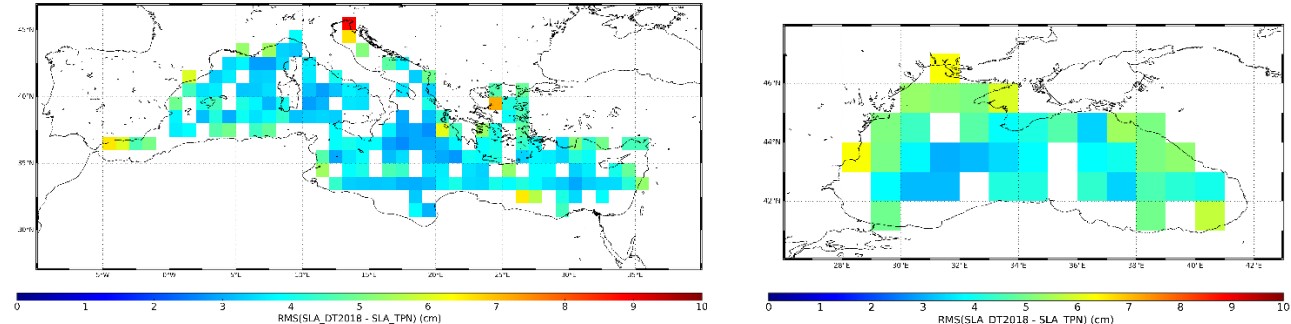

**Figure 9: RMS of the difference between regional Mediterranean Sea (left frame) and regional Black Sea (right frame) gridded DUACS DT-2018 sea level anomaly and independent TP along-track measurements over the period [2003-2004] (units: cm). The histogram above the colorbar indicates the number of occurrences of each value in the RMS map.**




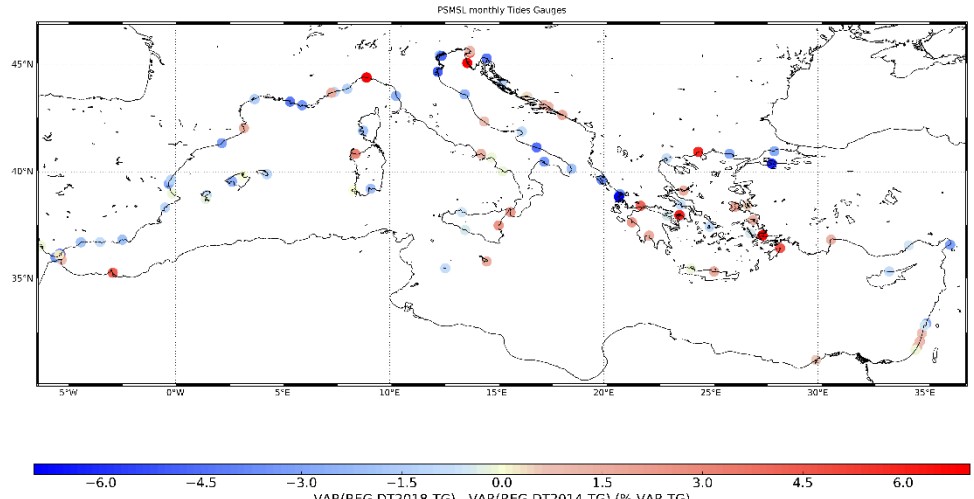

**Figure 10: Difference of the variance of the altimeter SLA minus TG SLA differences, using successively DT2018 and DT2014 SLA regional Mediterranean gridded products. Monthly Tide Gauges come from PSMSL network. Negative values mean that the SLA differences between altimetry and TGs are reduced when considering DT2018 regional Mediterranean gridded products.**







**Figure 11: (a) Map of the Mediterranean Sea showing the geographical limits and the nomenclatures of the regions (blue boxes) as defined in Manca et al. (2004) where drifter data is available in the western sub-basin: Alboran Sea (DS1), Balearic Sea (DS2), western and eastern Algerian (DS3 and DS4), Algero-Provençal (DF1), Liguro-Provençal (DF3, DF4), Gulf of Lion (DF2),Tyrrhenian Sea (DT4), Sardinian channel (DI1), Tyrrhenian Sea (DT2, DT3) and Sicily Strait (DI3); and in the eastern sub-**





basin: Adriatic Sea (DJ1, DJ2, DJ3), Ionian Sea (DJ4, DJ5, DJ6, DJ7, DJ8), Aegean Sea (DH1, DH2), Cretan Passage (DH3) and Levantine basin (DL1, DL2, DL3, DL4). Left column: maps of the Mediterranean Sea showing the comparison between DT-2018 regional altimetry product with the drifter in-situ observations within the geographical limits and the nomenclatures of the regions defined in (a). The statistical parameters showed are: (b) RMS difference; (c) correlation coefficient; (d) altimetry variance and (e) drifter variance. Right column: improvements (%) of the comparisons between the DT-2018 regional product and drifter in-situ observations with respect to the comparisons by using the DT-2014 product within the geographical limits and the nomenclatures of the regions defined in (a). The statistical parameters showed are: (f) RMS difference; (g) correlation coefficient and (h) altimetry variance. Positive values denote an improvement of DT-2018 regional product over DT-2014.