# Peer review of "DUACS DT2018: 25 years of reprocessed sea level altimetry products"

_Ocean Science, 2018_

## Referee Comment (RC1) · Anonymous Referee #1 · 18 Feb 2019

General comments Sorry, but I ran out of energy before finishing reading this paper. I found it to be extremely superficial, repetitive and unclear. I think it would be very difficult for altimetry beginners to understand, and too vague to be informative for experts, so I'm not sure what audience it would be useful for. So I suggest it goes back for a major rewrite, and well as addition of more information.

Specific comments (written as I read the paper) Abstract 1) "new altimeter standards ... has been used" 1) what are 'altimeter standards'? I think I know but many people won't. Especially in an Abstract, please use language that people will understand. 2) change 'has' to 'have'.

Intro 1) Sentence 1: "so called" -> "called". "Exists" -> "has existed"

[Figure]

2) p2 line 1 focus->focusses. I think I'll stop noting grammar edits. There are too many.

3) p1 l5 "Sentinel-3 L3 products are processed on behalf of EUMETSAT". This is confusing for some readers on 2 counts: they might not know what Sentinel-3 means. You just have to say "the Sentinel-3 altimeter mission". Don't use the passive verb "are processed", especially straight after saying that who does the work has just changed. Say who now does it.

4) "standards" see above

5) line 20. "standards". That words again, but this time I start to think I don't know what is meant. "processing from the standards to L3 and L4 products". This is terminology that is common among remote sensing specialists but is unfamiliar to a large fraction of the target audience. The previous paragraph referred to two products in meaningful language. Connect back to those products now via simple names. I don't think those are L3 and L4 but I might be wrong.

Data Processing 1) "cumulated" I don't think this is a real word. I think you mean "26 mission-years". Ie the sum of all the mission durations. This term was used before but I let it slip.

2.1 2) "complementary" this is very vague. If you are going to mention HY2A and its problems there is no point being cryptic and making people guess what you mean.

2.2 3) "geophysical standards". OK this is where we define what was referred to earlier as "altimeter standards and geophysical corrections". I see now why you have chosen a nice simple term like 'standard' but I'm sorry, I think it is too meaningless to be useful. I know this debate is old but I think this solution is a very bad one. New users will be confused by it. I think you need a quick little explanation explaining the equation SLA=Range-range_corrections-orbit-MSS-HF_alias_terms, noting that the terms in that equation are not really 'corrections'. The altimeter measures what it measures, which is not quite what everyone wants, for all purposes. De-tiding is not making the

answer more correct. It is making it wrong if you want the tide still there. Similarly for DAC. It is only for the purpose of making gridded SLA products that all these terms are needed, so start by saying that.

4) Table 1 columns are variable-span. Ie some entries span several columns but it is not clear which. I'm not sure all the entries are defined, either. E.g. I can imagine people wondering what a GDR-E orbit is.

5) "FES2014 is the last version" I think you mean 'latest' - except that's wrong I believe. FES2015?

2.3 1) "homogenise" this is cryptic for most readers. I think you mean that the non-Jason missions are debiased, taking Jason-class missions as 'truth' (once debiased, which is another thing to explain).

2) "...expose major changes that occurred in this DT2018 version. For an advanced description of the DUACS processing, readers are advised to consult Pujol et al., 2016. Say this earlier. However, see the next comment.

2.3.1 1) lines 25-33 "the cross-calibration step...." I see no mention of a change, so maybe this text can be shortened at lot (if this document is only about changes, as above).

2.3.2 1) "The along-track generation for repetitive altimeter mission is based on the use of a mean profile (MP) (Dibarboure et al., 2011 and Pujol et al., 2016). These MPs are necessary to co-locate sea surface heights of the repetitive tracks and to retrieve a precise mean reference for the computation of sea level anomalies. The methodology used for the DT2018 MP computation is the same as in DT2014." This is a perfect example of a sentence that I see no audience for. 'Experts' know this already. Beginners won't understand it: it is too unclear. Finally, it says there is no change since DT2014, contradicting 2.3 comment 2).

2)"For non-repetitive missions (ERS-1 during its geodetic phase, Cryosat-2, Hayaing-

2A, Jason-1 geodetic phase, Jason-2 geodetic phase, Saral-AltiKa geodetic phase), no MP can be estimated. The SLA is then derived along the real altimeter tracks using the gridded MSS." same comment as above. You need to either clearly explain the difference between MP and MSS, or assume it is understood.

2.3.3 lines 1-20. This is very uninformative. 'updated' and 'refined' are very uninteresting to read.

2.4 lines 23-32: this is just repetition of what was said earlier in this paper. Nor is it anything new. It is well known. I'm startng to lose my patience with this paper now.

lines 11-13: "As a second difference, the reference used to compute the Sea Level Anomalies is a Mean Sea Surface (MSS) for all missions in the C3S products whereas a mean profile of sea surface heights is used...." Back to this issue again. Very confusing. See comment 2 on 2.3.2 above.

—to end of 2.4. As far as I can tell, this is all old information that experts don't need to be told, and beginners won't understand, the way it is described here.

Section 3. 1) Results section. But I feel unready to read about results. All I have gleaned so far is that some updates have been made, with very few details given.

2) "Additional variance is observed for high variability regions in DT2018 products and is linked to the new OI parametrization." 'linked' is it? I'm getting more and more annoyed about this persistent abscence of information. Is it secret?

3) p8 line 4-5: "At high latitude, the difference of variance is important ($100cm^2$ to $200cm^2$) and is linked to the new MSS correction." It's not obvious to me how this could be true. It must be a fairly convoluted argument.

4) p8 line 11-12: "However, in the equatorial band ($\pm20°$N), the EKE in the DT2018 is less important (-17%). This is linked with the evolution of the noise measurement considered in the mapping process for all satellites." I'm getting really sick of this vague uninformative style: 'linked' and 'evolution'.

5) p8 line 19-29: Discussion of table 3. This is an important part of this study, but lots of information is missing. Table 3 has just 2 values for each of 4 regions. Why trim it down to such a bare minimum of information? E.g. For the reference area |track-map|^2=1.4cm^2. This is for a 'low variability' region. But how low? Easy to answer: list the |track|^2 and |map|^2 values as well.

6) p8 line 19-29: Discussion of table 3. —-also: this is just for the 2-sat product. What about the multisat product? I hear the answer already: "Because none of the data are withheld". My response: this does not stop you listing the map minus track stats, which are then measures of the closeness of fit (as distinct from map error). To estimate map error, pick a time with many good satellites and rerun the OI, withholding one (e.g. C2) for use as the error measurer.

7) p9 line 1-2 "Positions and velocities of drifters are interpolated using a 3-day low-pass filter in order to remove high-frequency motions." I have 3 grumbles: i) don't use the passive voice ('are interpolated') - it leaves it to the reader to guess who did the interpolating - we assume it was you but we can't be sure. ii) this is a very brief 'Methods' section squeezed into the Results section iii) why remove 'high frequency motions?' A 3-day filter also removes a lot of low-frequency Eulerian velocity (a drifter can easily go 1/4 of the way around a well-resolved eddy in 3 days). So, instead of filtering then differencing, it is better to do differencing then filtering.

8) Fig. 6: It seems to me that 2 panels are missing: the ones showing the DT2018-DT2014 difference.

9) p9 line 4-5 "the comparison reveals that DT2018 altimetry products underestimate absolute geostrophic current." This statement is not supported by Fig. 6, Table 4, or by the mention that someone (we don't know who, because passive verb was used) has done a Taylor diagram (but kept the results to themselves - all we know is that the results are 'strong'). As in comment 5 above, list the variance of the drifter and altimetric velocities in order to prove that the altimetry under-estimates the drifter velocities.

[Figure]

10) p9 line 10-17. This discussion only talks (vaguely, but I'm not going to mention this any more because it is everywhere) about DT2018 being better DT2018, which is good news, but what people really want to know is the error:signal ratio.

3.3 1) p9 line 19-33. This is all repetition.

2) p10 line 1-10. This is an interesting result that is "not understood yet". I think you could try a little harder. I see red dots (DT2018 is worse) on W and E USA, Spain (as mentioned) but also Japan - all 30-45N. Let's see some example time-series of errors for each product individually, not to mention the two signals being differenced (altim and TG) individually as well.

3.4-onwards

Sorry, but I am not prepared to read any further. I think this paper has too many faults to be published in close to its present form.
* * *

---

## Short Comment (SC1) · 6 Mar 2019

This paper presents findings from assessment of the quality of the DT-2018 products versus DT-2014. I find that the most convincing improvement is near coast and in the Med Sea and the Black Sea. The interpretation of the open ocean performance is not compelling. The following are some specific comments:

P.1 Introduction- I'd suggest adding some text on the history of altimetry missions over the past 25 years.

p.2 last line- Is the data from Hayaing-2 A incorporated in DT2018?

p.3 first line- What about the data distribution by NASA? Line 6- is the altimetry community represented by the OSTST? If so, please mention it.

p.5 first line- cite Table 2 when the mean period (MP) is first introduced. Line 5- what is "upstream measurements"? Line 17- give a reference for the MSS. Line 18 - delete "of" after "benefit" Line 24 - What is "Theoretical Track"?

p.6 line 11-give reference for the MSS

p. 7 line 16- delete "at" after "be" line 29-30 - Is "additional variance for high variability regions in DT2018" an improvement? if so, why?

p.8 line4- why is the difference of variance important? What does it mean? Line 9- How is the EKE at the equator computed while geostrophy breaks down there? Line 11- What does it mean by "less important"? Line 16- Given the issue of geostrophy near the equator, how would one interpret the equatorial EKE reduction as improvement?

p.9 line 4- Is the fact that DT2018 products underestimate absolute geostrophic current an improvement? If not, what is the interpretation? line 5- The equatorial regions in Fig 6 are blocked but not in Fig 5? line 13 - What does it mean by "improvement is clearly visible in the intra-tropical band" while the regions are blocked in Fig 6?

p.10 line7- Please quantify the global reduction of the variance. line15 - What are the "three estimates"? I see only two in Fig 8 left.

p.11 lines 13-15- I think the information of Table 5 is sufficient and Fig 9 can be deleted. It does not convey much additional information. Line 26- Please quantify the overall improvement shown in Fig 10.

---

## Referee Comment (RC3) · Anonymous Referee #3 · 26 Mar 2019

General Comment :

\*\*\*\*\*\*\*\*\*\*\*\*\*\*\*\*

The manuscript presents the overall enhancement of gridded and along-track altimetry products following the DT2018 reprocessing, in a way that is similar to the DT2014 reassessment published earlier. Methods and Processing for quality assessment are therefore established, and skill assessment has not been developped further, but this is acceptable to me. I believe it is a necessary step to publish such reassesment peridodically, and to synthetize skill metrics for the state-of-the-art altimetry products as proposed. I therefore support the publication of this manuscript, suggesting some modifications below. Title is appropiate.

[Figure]

* As a suggestion : I believe the whole manuscript could be summarized on a single figure, in the form of a target or taylor diagram showing skill metrics for the different products (along-track, gridded SLA, gesotrophic currents) and scales (regional, global coastal, global offshore, climatic, etc ..) showing DT2014 postions and DT2018 positions. This is a mere suggestion, but I think it would provide a very efficient overview of the DT2018 update. Unless there are good justifications why this can not be done (at least for part of the datasets presented), I think it would be relevant for the manuscript to consider issuing this figure.

Specific Comments (I start with question mark "?" to denote a suggestion)
* * *
* Abstract: P1L19 : I understand the reason for providing quantitative metrics in the abstract, but the term "errors" is too vague in the present abstract. Please precise.

* Text :

P3L5-6 :? recommendationS, correctionS

P3L33: "in Deep Ocean" -> "in the deep ocean".

P4L18 : It would ease the read to define "geoditic" and "drifting" mission, and help non-specialized readers to grasp the challenges of altimetry processing.

P4L23 : please define more clearly the "percentage of data recovery"

P5l20 : complete: differences of ...

P5l29 "law-pass" -> "low-pass"

P6L6:7: ? consider Capet et al. 2014 that adressed those issue for DT2014.

P6L9 : Does "selection" applies on 1) altimeter data for along-track data product generation or 2) along-track product for gridded products generation ?

P6L14:15 vs P6L20:21 : There seems to be apparent contradictions here, please
rephrase for clarity ( ".. unchanged for global and Black Sea, wrt to DT2014" VS "BlackSea paramters are NOW similar to global, except for scales ... ".

P6L27: correct "Different parameters leadS"

P7L30: There is a problem in the sentence "This ... variance". Even after displacing "the", the meaning is not clear, please clarify.

P8L4: precise the sign of the 100-200 cm$^2$ difference of variance (but I think it's both plus and minus).

P8L17 : rephrase "less peaky"

P8L22 : could you explain why only th period 2003-2004 can be considered for this assessment ?

P8L23: The author avoided the nomenclature "two-sat"/"all-sat" up to this point. Can it be also avoided here ? (I think it is the only place where it is used).

P9L8 : ? is it "COvariance and RMS" ?

P9L10 : "altimeter maps" -> "geostrophic current maps"

P9L12 : lowercase "Variance"

P9L20 "points" -> "data points"

P9L20/22 : rephrase "We gain all points".

P9L26 "in the" repeated

P10L4 : Why "maximum" correlation ? Does that refer to a selection amongst the neighboring pixels ?

p10L26 : "a measurementS"

P11L3, remove "." after "yr" (2x).

P11L18 "For" -> "for"

P11L26:28 Why is there no TG validation for the Black Sea ? Explain.

P12L14 "large" -> "largeR"

p12l22 "lager" -> "larger"

P13L8 "for" -> "from"

P13L26 Biblio ref for eddy tracking, instead of html ?

* Figures & Tables :
* * *
* Are appropriated and all useful in general.

* Small to very small coordinates, axes and colorbar title. Please ensure readability.

Fig 1: What determines the end of the bars for the future ? scheduled lifetime ? please precise.

Fig 2: Probably the less useful figure. If considered essential, should the figgure be re-processed with larger bins ? It does not provides many information as for now, except : "more data in the 20km coastal band", "lot of noise in the center" and " a strange, un-commented blue track in the center of East Med". Unless justified otherwise, i suggest to remove this figure.

Fig 3,: caption : rephrase "Loss ones".

Fig 6. Second half of the caption ("Difference of the variance ... ". Does not correspond to the figure (eg. refers to negative values). -> ? missing panel ?

Fig 9: Caption mentions histograms that are not visible on the figure.

Fig 10 : use divergent colormap for the panel f,g,h (eg. blue-white-red)

\* References :

\*\*\*\*\*\*\*\*\*\*\*\*

\* There are many references to work 'in prep.', including on to "In prep. to be submitted to OD in 2016" (Lyard et al.) . Please check with editorial office on the policy as regards reference to unpublished works.

\* The reference style is not homogeneous, with years being given some times at the end, some times after the authors. Please homogenize.

\* There are (many) reference works not provided in the bibliography (eg. Valladeau et al, 201 ; Le Traon et al, 1998, Ducet et al 2000, Le Traon & Ogor 1998 ; Le Traon et al, 2003 ; Lumpkin et al. 2013 ; Taylor, 2001 ; Watson et al, 2015 ; Beckley et al , 2017 ; Dieng et al 2017; Ballarota, in prep ; d'Ovidio 2015.)

\* Similarly there are (many) references in the biblio that are not mentionned in the text. I do not think it is my duty to revise this for you extensively. Please check carefully.

---

## Author Comment (AC1) · 28 May 2019

Authors: We acknowledge Rev. #1 for his/her review. All comments and remarks have been considered. In the next paragraphs we present the reviewer's comments followed by our point-by-point reply (blue color).

General comments: Sorry, but I ran out of energy before finishing reading this paper. I found it to be extremely superficial, repetitive and unclear. I think it would be very difficult for altimetry beginners to understand, and too vague to be informative for experts, so I'm not sure what audience it would be useful for. So I suggest it goes back for a major rewrite, and well as addition of more information.

Authors: Considering your comments, we have tried to make the necessary modifications to improve the manuscript. However, this article is not intended to provide a course on altimetry processes for beginners but rather to present a new dataset. The structure and organization of this article was intended to be very similar to what had been done in the article dedicated to the DUACS DT2014 dataset (Pujol et al., 2016). We realized that some sections deserved important clarifications. We added them to the new version of the manuscript.

Specific comments (written as I read the paper) Abstract 1) "new altimeter standards ... has been used" 1) what are 'altimeter standards'? I think I know but many people won't. Especially in an Abstract, please use language that people will understand. 2) change 'has' to 'have'.

Authors: The term "altimetry standards" regroups algorithms and parameters used to estimate the different fields of the equation: SLA=orbit-range-∑correction-MSS. The notion of "altimeter standards" or "standards" is commonly used in the literature and particularly in the last two articles concerning DUACS reprocessing: Dibarboure et al. (2010) and Pujol et al. (2016). The current manuscript being closely linked to these two papers we have chosen to keep this notion. A dedicated chapter of the manuscript entitled "Altimetry standards" presents and explains in detail what these standards correspond to (section 2.2 Altimetry standards).

As recommended by the reviewer (see also comments 4) 5) and 2.2 3)) and for greater clarity, we have added details to the specific chapter « 2.2 Altimetry standards ».

Intro 1) Sentence 1: "so called" -> "called". "Exists" -> "has existed"

2) p2 line 1 focus->focusses. I think I'll stop noting grammar edits. There are too many.

Authors: The authors have asked for an English grammar and spelling correction service in order to improve the quality of the manuscript.

3) p1 l5 "Sentinel-3 L3 products are processed on behalf of EUMETSAT". This is confusing for some readers on 2 counts: they might not know what Sentinel-3 means. You just have to say "the Sentinel-3 altimeter mission". Don't use the passive verb "are processed", especially straight after saying that who does the work has just changed. Say who now does it.

Authors: Done.

4) "standards" see above

Authors: See above

5) line 20. "standards". That words again, but this time I start to think I don't know what is meant. "processing from the standards to L3 and L4 products". This is terminology that is common among remote sensing specialists but is unfamiliar to a large fraction of the target audience. The previous paragraph referred to two products in meaningful language. Connect back to those products now via simple names. I don't think those are L3 and L4 but I might be wrong.

Authors:  The sentence has been rewritten for more clarity.

Data Processing 1) "cumulated" I don't think this is a real word. I think you mean "26 mission-years". Ie the sum of all the mission durations. This term was used before but I let it slip.

Authors: Done.

2.1 2) "complementary" this is very vague. If you are going to mention HY2A and its problems there is no point being cryptic and making people guess what you mean.

Authors: This section has been rewritten.

2.2 3) "geophysical standards". OK this is where we define what was referred to earlier as "altimeter standards and geophysical corrections". I see now why you have chosen a nice simple term like 'standard' but I'm sorry, I think it is too meaningless to be useful. I know this debate is old but I think this solution is a very bad one. New users will be confused by it. I think you need a quick little explanation explaining the equation SLA=Range-range_corrections-orbit-MSS-HF_alias_terms, noting that the terms in that equation are not really 'corrections'. The altimeter measures what it measures, which is not quite what everyone wants, for all purposes. De-tiding is not making the answer more correct. It is making it wrong if you want the tide still there. Similarly for DAC. It is only for the purpose of making gridded SLA products that all these terms are needed, so start by saying that.

Authors: See the discussion above concerning the term "altimetry standards". This section has been rewritten. In addition, a paragraph concerning specific along-track (L3) products has been added in section 2.4. It introduces the possibility to remove specific geophysical effects that are taken into account in the DUACS processing.

This article is not intended to provide a course on altimetric processes for beginners but rather to present a new dataset. The readers are advised to refer to the existing literature presenting the altimeter measurements. We have added a specific reference: Escudier, P., Couhert, A., Mercier, F., Mallet, A., Thibaut, P., Tran, N., Amarouche, L., Picard, B., Carrère, L., Dibarboure, G., Ablain, M., Richard, J., Steunou, N., Dubois, P., Rio, M. H., and Dorandeu, J.: Satellite radar altimetry: principle, geophysical correction and orbit, accuracy and precision, in: Satellite Altimetry Over Oceans and Land Surfaces, edited by: Stammer, D. and Cazenave, A., CRC Press, Taylor & Francis, Boca Raton, 2018, https://doi.org/10.1201/9781315151779 and in particular section 1.6.2 (tides, high frequency signals).

4) Table 1 columns are variable-span. Ie some entries span several columns but it is not clear which. I'm not sure all the entries are defined, either. E.g. I can imagine people wondering what a GDR-E orbit is.

Authors: The authors have used the Copernicus Publications Word template to create Table 1. However, we have made it evolve for a better readability. This new format remains to be discussed with the publisher. In the line corresponding to the orbit parameter, the GDR mention has been replaced by POE. Indeed, Geophysical Data Record (GDR) corresponds to the generic term for L2 altimeter product whereas POE (Precise Orbit Estimation) is the exact and appropriate term. The acronym DAC have also been clarified: Dynamic Atmospheric Correction.

5) "FES2014 is the last version" I think you mean 'latest' - except that's wrong I believe. FES2015?

Authors: At the time the DT2018 products were computed, FES2014 was the latest version available. This preliminary version, noted FES2014a, has been produced in 2015 based on GOT4v8ac loading tide. Then new tide loading effects have been computed using FES2014a oceanic tide. These FES2014a tide loading effects have been used to produce the final model version noted FES2014b.

2.3 1) "homogenise" this is cryptic for most readers. I think you mean that the nonJason missions are debiased, taking Jason-class missions as 'truth' (once debiased, which is another thing to explain).

Authors: «homogenise » is used page 3 line 25 as an introduction of two different processes that are described in the following sections: global and regional bias reduction to ensure mean sea level stability and cross-calibration process to minimize inter-missions' errors at crossover. For a complete description of the processes, the authors explicitly guide the reader to a much more detailed reference: Pujol et al., 2016.

2) "...expose major changes that occurred in this DT2018 version. For an advanced description of the DUACS processing, readers are advised to consult Pujol et al., 2016. Say this earlier. However, see the next comment.

Authors: This is a reminder of the approach (see. p2L1) explaining that this article focuses on improvements of the DT2018 dataset compared to the DT2014. Thus, we think that it is adapted to keep these sentences in the introduction of section 2.3 "Evolution of the DUACS processing".

2.3.1 1) lines 25-33 "the cross-calibration step...." I see no mention of a change, so maybe this text can be shortened at lot (if this document is only about changes, as above).

Authors: Done.

2.3.2 1) "The along-track generation for repetitive altimeter mission is based on the use of a mean profile (MP) (Dibarboure et al., 2011 and Pujol et al., 2016). These MPs are necessary to co-locate sea surface heights of the repetitive tracks and to retrieve a precise mean reference for the computation of sea level anomalies. The methodology used for the DT2018 MP computation is the same as in DT2014." This is a perfect example of a sentence that I see no audience for. 'Experts' know this already. Beginners won't understand it: it is too unclear. Finally, it says there is no change since DT2014, contradicting 2.3 comment 2).

Authors: There is indeed no change in methodology (this is why the two references to Pujol et *al.*, 2016 & Dibarboure et *al.*, in review are mentioned) but the data selection has evolved (from line 5). Thus, the authors think that it is appropriate to briefly recall the interest of mean profiles without going into details. They mention references that are relevant for the uninitiated readers. The authors added a reference that precisely details the usefulness and processing of MP (Dibarboure et *al.*, in review). To facilitate the understanding, we considered appropriate to retain the short sentence "These MPs are necessary to co-locate sea surface heights of the repetitive tracks and to retrieve a precise mean reference for the computation of sea level anomalies".

2)"For non-repetitive missions (ERS-1 during its geodetic phase, Cryosat-2, Hayaing 2A, Jason-1 geodetic phase, Jason-2 geodetic phase, Saral-AltiKa geodetic phase), no MP can be estimated. The SLA is then derived along the real altimeter tracks using the gridded MSS." same comment as above. You need to either clearly explain the difference between MP and MSS, or assume it is understood.

Authors: The authors have chosen to keep the sentence to facilitate the understanding of the following paragraph about the MSS (2.3.3 L11). Nevertheless, and as suggested, we have added references (Pujol et al., 2016 and Dibarboure et al., in review) which can help the user to have access to more details.

2.3.3 lines 1-20. This is very uninformative. 'updated' and 'refined' are very uninteresting to read.

Authors:  This section lacked details; we have enriched it. The words "updated" and "refined" have been deleted and replaced by more precise descriptions of the developments implemented.

2.4 lines 23-32: this is just repetition of what was said earlier in this paper. Nor is it anything new. It is well known. I'm startng to lose my patience with this paper now.

Authors: These lines are indeed redundant with the explanations given in the introduction. This has been simplified (p2 l11-19).

lines 11-13: "As a second difference, the reference used to compute the Sea Level Anomalies is a Mean Sea Surface (MSS) for all missions in the C3S products whereas a mean profile of sea surface heights is used...." Back to this issue again. Very confusing. See comment 2 on 2.3.2 above.

—to end of 2.4. As far as I can tell, this is all old information that experts don't need to be told, and beginners won't understand, the way it is described here.

Authors: According to the authors, this major difference between CMEMS and C3S products has never been addressed (and should thus not be considered as old information) and must be described to expose the specificities of the different Copernicus products.

The product dedicated to climate applications (C3S) is based on a stable number of missions (two) in the satellite constellation and has a specific processing (which is the interest of section 2.4 and particularly from line 11 to end), that follows the recommendation made within external R&D projects (such as the ESA Sea Level Climate Change Initiative project). Along-track data were not calculated with a MP but only with the MSS (and even for repetitive missions) which contributes to improve the mean sea level stability (especially for regional products). Thus, this should not be considered as "old information", since this has been implemented for the recent production of the C3S sea level products.

Section 3. 1) Results section. But I feel unready to read about results. All I have gleaned so far is that some updates have been made, with very few details given.

2) "Additional variance is observed for high variability regions in DT2018 products and is linked to the new OI parametrization." 'linked' is it? I'm getting more and more annoyed about this persistent abscence of information. Is it secret?

Authors: The wording of the sentence has been changed and details added.

3) p8 line 4-5: "At high latitude, the difference of variance is important (100cm2 to 200cm2 ) and is linked to the new MSS correction." It's not obvious to me how this could be true. It must be a fairly convoluted argument.

Authors: Pujol et *al.*,2018 shows the new MSS15 is more extended at high latitude than the old one. (see also figure 1 below). This allows us to compute the OI with much more precision and stability in this region. The figure 4 of Pujol et *al.*,2018 shows the difference of the variance of SLA along HY2A tracks. These differences are major at high latitude.

Figure 2 (below) shows the difference of SLA variance with DT2018 and DT2014 gridded products from the same point of view as figure 1. The difference in spatial coverage of the two MSS explains the difference in quality of the SLA grid products in this area.

[Figure]

*Figure 1: Mean Sea Surface CNES CLS 2011 version (left panel) and MSS CNES CLS 2015 version (right panel).*

[Figure]

*Figure 2: Difference between SLA variance observed with DT2018 and DT2014 gridded products. Same figure as figure 4 of the manuscript but centered on the North Pole. Units: cm².*

4) p8 line 11-12: "However, in the equatorial band (±20∘N), the EKE in the DT2018 is less important (-17%). This is linked with the evolution of the noise measurement considered in the mapping process for all satellites." I'm getting really sick of this vague uninformative style: 'linked' and 'evolution'.

Authors: The sentence has been changed and details added.

5) p8 line 19-29: Discussion of table 3. This is an important part of this study, but lots of information is missing. Table 3 has just 2 values for each of 4 regions. Why trim it down to such a bare minimum of information? E.g. For the reference area |trackmap|^2=1.4cm^2. This is for a 'low variability' region. But how low? Easy to answer: list the |track|^2 and |map|^2 values as well.

Authors: The low variability region has been introduced in Pujol *et al.,*2016. The authors found interesting to reuse it to have a reference area where observations errors are small. The SLA variability in this region is between 0 and 7 cm². This precision has been added to the Table 3. A figure (figure 5 in the new manuscript version) has been added in the manuscript to show the RMS difference (in % of RMS) between two-sat gridded products and along-track product for DT2018 and DT2014 versions.

We also added a discussion about improvements in the intertropical zone.

6) p8 line 19-29: Discussion of table 3. —-also: this is just for the 2-sat product. What about the multisat product? I hear the answer already: "Because none of the data are withheld". My response: this does not stop you listing the map minus track stats, which are then measures of the closeness of fit (as distinct from map error). To estimate map error, pick a time with many good satellites and rerun the OI, withholding one (e.g. C2) for use as the error measurer.

Authors: This issue has been discussed p8 between line 23 to 25. The error described here must be considered as the upper limit. We choose not to describe in the manuscript a configuration with more than two satellites. However, the authors also studied the period 2016-2017, and the

conclusions are similar with C2 as an independent along-track mission. (using Jason-2 and AltiKa for the mapping process).

The L4 all-sat validation is complemented by *in situ* drifter's comparison.

7) p9 line 1-2 "Positions and velocities of drifters are interpolated using a 3-day lowpass filter in order to remove high-frequency motions." I have 3 grumbles: i) don't use the passive voice ('are interpolated') - it leaves it to the reader to guess who did the interpolating - we assume it was you but we can't be sure. ii) this is a very brief 'Methods' section squeezed into the Results section iii) why remove 'high frequency motions?' A 3-day filter also removes a lot of low-frequency Eulerian velocity (a drifter can easily go 1/4 of the way around a well-resolved eddy in 3 days). So, instead of filtering then differencing, it is better to do differencing then filtering.

Authors: The authors added a relevant reference which explain the interest and the method used for 3-day lowpass filtering: Use of Altimeter and Wind Data to Detect the Anomalous Loss of SVP-Type Drifter's Drogue M.-H. Rio. 2012.
The main objective of the filtering process is to discard the tide and the inertia in drifters' data.
We know that: - we don't filter enough between 10S and 10N to get rid of all the inertia
            - we filter a little too much at high latitudes, knowing that we don't want to go below
        24 days for the tide.
The 3-day period is a compromise between these two. The methodology still needs to be improved.

8) Fig. 6: It seems to me that 2 panels are missing: the ones showing the DT2018- DT2014 difference.

Authors: The authors have added the missing plot and related comments.

9) p9 line 4-5 "the comparison reveals that DT2018 altimetry products underestimate absolute geostrophic current." This statement is not supported by Fig. 6, Table 4, or by the mention that someone (we don't know who, because passive verb was used) has done a Taylor diagram (but kept the results to themselves - all we know is that the results are 'strong'). As in comment 5 above, list the variance of the drifter and altimetric velocities in order to prove that the altimetry under-estimates the drifter velocities.

Authors: The authors modified the sentence. It is neither an improvement nor a degradation of the products' quality but it is rather described as it is. It was also noted by Pujol et al,2016 in the DT2014 version of the sea level products.

The authors also added the RMS difference between gridded and independent drifters' measurements for DT2018 and DT2014. Related comments have been added.

10) p9 line 10-17. This discussion only talks (vaguely, but I'm not going to mention this any more because it is everywhere) about DT2018 being better DT2018, which is good news, but what people really want to know is the error:signal ratio.

Authors: The error is estimated using independent data for the SLA and geostrophic current on high variability and low variability region, coastal areas… (Table 3 to 6). The authors do not see what additional information could be added.

3.3 1) p9 line 19-33. This is all repetition.

Authors: The authors have streamlined this section.

2) p10 line 1-10. This is an interesting result that is "not understood yet". I think you could try a little harder. I see red dots (DT2018 is worse) on W and E USA, Spain (as mentioned) but also Japan - all

30-45N. Let's see some example time-series of errors for each product individually, not to mention the two signals being differenced (altim and TG) individually as well.

Authors: We know from Saraceno et al, 2018 (Estimates of sea surface height and near-surface alongshore coastal currents from combinations of altimeters and tide gauges) that coastal processes are more difficult to resolve with altimeter data, because of two types of problems. First, and most importantly, intrinsic difficulties affect the corrections applied to the altimeter data near the coast (e.g., the wet tropospheric component, high-frequency oceanographic signals, tidal corrections, etc.). Thus, data are usually flagged as unreliable within some distance of the coast. Second, the interpolation of along-track data collected by just one or two satellites provides only marginal resolution of mesoscale and smaller-scale structure in ocean circulation [Le Traon and Dibarboure, 2002; Leeuwenburgh and Stammer, 2002; Chelton and Schlax, 2003], which is dominant in the coastal region.

We did compare some time series for tide gauges on the Portuguese coast. It is difficult to draw conclusions about a particular time period over which comparisons are degraded. We were unable to correlate these degradations with periods when there are fewer data (fewer satellites in the constellation, or anomaly on a satellite).

We know that the new tide correction is particularly important in coastal areas, but again we have not been able to explain these degradations with this correction.

We are not in a position to explain the degradation observed in these well-located areas of the globe (West Coast of the USA, Portuguese coast, etc.).

3.4-onwards

Sorry, but I am not prepared to read any further. I think this paper has too many faults to be published in close to its present form.

---

## Author Comment (AC2) · 28 May 2019

**Fu Lee Lueng (Referee)** lee-lueng.fu@jpl.nasa.gov

Authors: We warmly acknowledge Lee Lueng Fu for his review. All comments and remarks have been considered. In the next paragraphs we present the reviewer's comments followed by our point-by-point reply.

This paper presents findings from assessment of the quality of the DT-2018 products versus DT-2014. I find that the most convincing improvement is near coast and in the Med Sea and the Black Sea. The interpretation of the open ocean performance is not compelling. The following are some specific comments:

P.1 Introduction- I'd suggest adding some text on the history of altimetry missions over the past 25 years.

Authors: Done

p.2 last line- Is the data from Hayaing-2 A incorporated in DT2018?

Authors: As shown in Figure1, Hayaing-2 A data are incorporated in DT2018. The particularity of the reprocessing is to integrate additional HY2A data that were not taken into account in the DT2014 production: data from March 2016 to February 2017. This paragraph has been rewritten to be more explicit.

p.3 first line- What about the data distribution by NASA?

Authors: L2P data are only distributed by CNES and EUMETSAT. The data distribution by agencies NASA, NSOAS, ISRO, ESA, EUMETSAT, CNES… are taken into account in L2 products. DUACS processing only uses L2P data. The sentence has been reformulated in the manuscript.

Line 6- is the altimetry community represented by the OSTST? If so, please mention it.

Authors: Done

p.5 first line- cite Table 2 when the mean period (MP) is first introduced.

Authors: Done

Line 5- what is "upstream measurements"?

Authors: These "upstream measurements" correspond to the L2P products that have been presented previously. The sentence has been rewritten.

Line 17- give a reference for the MSS.

Authors: Done.

Line 18 - delete "of" after "benefit"

Authors: Done.

Line 24 - What is "Theoretical Track"?

Authors: The authors added a reference (Dibarboure et *al.*, 2011) which provide appropriate details: "Altimetry satellites generally use repetitive orbits: after 10–35 days, the sensor flies over the same locations, hence the notion of cycles (time needed to revisit the same location) and the ability to co-locate data. However, the satellite ground track cannot be perfectly controlled and is kept only in a band about 1 km wide. It is thus necessary to use an arbitrary and mission-consistent position for the co-location process. SSH measurements are then projected onto these co-location points."

p.6 line 11-give reference for the MSS

Authors: Done.

p. 7 line 16- delete "at" after "be"

Authors: Done.

line 29-30 - Is "additional variance for high variability regions in DT2018" an improvement? if so, why?

Authors: At this stage, this diagnostic is only used to characterize the impact of the new mapping process and new altimeter corrections. It is not presented as an improvement (It might as well also correspond to noisier DT2018 products). The only conclusion is that there is more variability in DT2018 products. It is only in a second step, by comparing with independent dataset and *in-situ* measurements, that we show that this gain of variability corresponds to an improvement.

p.8 line4- why is the difference of variance important? What does it mean?

Authors: The authors have reformulated this sentence.

Line 9- How is the EKE at the equator computed while geostrophy breaks down there?

Authors: The geostrophic current products disseminated to users are computed using a nine-point stencil width methodology (Arbic et al., 2012) for latitudes outside the ±5°N band. In the equatorial belt, the Lagerloef methodology (Lagerloef et al,1999) introducing a $\beta$ plane approximation is used. The EKE is computed from this geostrophic estimation. This methodology did not changed since DT2014 version.

As at the equator the geostrophy breaks down, the ±5°N band is usually masked at the equator. Figure 5 has been corrected.

Line 11- What does it mean by "less important"?

Authors: The authors have reformulated this sentence.

Line 16- Given the issue of geostrophy near the equator, how would one interpret the equatorial EKE reduction as improvement?

Authors: The equatorial EKE reduction is a direct consequence of the increase of the noise measurements considered in the OI process: Observation errors have been increased in the equatorial belt, so the SLA signal is smoother and less energy is observed in this region. It has been noted that in DT2014 products, there was too much noise at the equator.

In the ±5°N band, near the equator, the EKE has been masked.

p.9 line 4- Is the fact that DT2018 products underestimate absolute geostrophic current an improvement? If not, what is the interpretation?

Authors: It is presented as a fact, not an improvement. Main reasons are that absolute geostrophic current from altimeter are smoother (fewer small scales) than with drifters, there is probably still some ageostrophic signal left in drifters' data.

line 5- The equatorial regions in Fig 6 are blocked but not in Fig 5?

Authors: The authors have corrected this mistake.

line 13 - What does it mean by "improvement is clearly visible in the intra-tropical band" while the regions are blocked in Fig 6?

Authors: The sentence has been reformulated to take into account that the ±5°N band is masked.

p.10 line7- Please quantify the global reduction of the variance.

Authors: Global reduction of the variance is around 0.6%. it has been added in the document.

line15 - What are the "three estimates"? I see only two in Fig 8 left.

Authors: The authors have reformulated the sentence. The first estimate using along-track measurements of the reference mission only (Ablain et al.,2017) is not display here.

p.11 lines 13-15- I think the information of Table 5 is sufficient and Fig 9 can be deleted. It does not convey much additional information.

Authors: The authors have replaced the figure with the difference of the root mean square of the SLA minus independent Tope/Poseidon along-track SLA, using successively DT2018 and DT2014 gridded product. The authors thought that the spatial information conveyed by this comparison would be more relevant. We have added a description of this new figure in the body of the manuscript.

Line 26- Please quantify the overall improvement shown in Fig 10.

Authors: Overall reduction of the variance for Mediterranean product is around 0.4%.

---

## Author Comment (AC3) · 28 May 2019

Authors: We warmly acknowledge Rev.#3 for his review. All comments and remarks have been considered. In the next paragraphs we present the reviewer's comments followed by our point-by-point reply.

General Comment :
* * *
The manuscript presents the overall enhancement of gridded and along-track altimetry products following the DT2018 reprocessing, in a way that is similar to the DT2014 reassessment published earlier. Methods and Processing for quality assessment are therefore established, and skill assessment has not been developped further, but this is acceptable to me. I believe it is a necessary step to publish such reassesment peridodically, and to synthetize skill metrics for the state-of-the-art altimetry products as proposed. I therefore support the publication of this manuscript, suggesting some modifications below. Title is appropiate.

* As a suggestion : I believe the whole manuscript could be summarized on a single figure, in the form of a target or taylor diagram showing skill metrics for the different products (along-track, gridded SLA, gesotrophic currents) and scales (regional, global coastal, global offshore, climatic, etc ..) showing DT2014 postions and DT2018 positions. This is a mere suggestion, but I think it would provide a very efficient overview of the DT2018 update. Unless there are good justifications why this can not be done (at least for part of the datasets presented), I think it would be relevant for the manuscript to consider issuing this figure. Specific Comments (I start with question mark "?" to denote a suggestion)

Authors: The authors do agree that this suggestion is a good idea. We have tried to compute such figure reusing existing results, and particularly Table 3 to 5. However, the result does not appear to us to be sufficiently successful to be published. It would deserve much more substantive work. The authors keep the idea and will try to implement it in future quality document associated with the DT2018 products and for future reprocessing.

[Figure]
* * *
* Abstract: P1L19 : I understand the reason for providing quantitative metrics in the abstract, but the term "errors" is too vague in the present abstract. Please precise.

Authors: The authors specified that these values have been computed using independent and *in-situ* measurements. In particular, the difference in variance of difference between altimetry and independent dataset allows to characterize this error.

* Text :

P3L5-6 :? recommendationS, correctionS

Authors: Done

P3L33: "in Deep Ocean" -> "in the deep ocean".

Authors: Done

P4L18 : It would ease the read to define "geoditic" and "drifting" mission, and help nonspecialized readers to grasp the challenges of altimetry processing.

Authors: The authors replaced the terms "geodetic" and "drifting" by "non-repetitive mission".

P4L23 : please define more clearly the "percentage of data recovery"

Authors: The authors have reformulated this sentence which was very confusing. There was no data in DT2014 products and now validated measurements are available.

P5l20 : complete: differences of ...

Authors: Difference of SLA. It has been specified both at line 19 and 20.

P5l29 "law-pass" -> "low-pass"

Authors: Done

P6L6:7: ? consider Capet et al. 2014 that adressed those issue for DT2014.

Authors: Indeed, this sentence is incorrect/misunderstood. The authors rewrote this passage taking into account the publication Capet et al., 2014.

P6L9 : Does "selection" applies on 1) altimeter data for along-track data product generation or 2) along-track product for gridded products generation ?

Authors: it is for gridded product generation. The explanation has been clarified.

P6L14:15 vs P6L20:21 : There seems to be apparent contradictions here, please C2 rephrase for clarity ( ".. unchanged for global and Black Sea, wrt to DT2014" VS "BlackSea paramters are NOW similar to global, except for scales ... ".

Authors: Done

P6L27: correct "Different parameters leadS"

Authors: Done

P7L30: There is a problem in the sentence "This ... variance". Even after displacing "the", the meaning is not clear, please clarify.

Authors: Additional variance, between 2% and 5%, is observed for high variability regions in DT2018 products.

P8L4: precise the sign of the 100-200 cm2 difference of variance (but I think it's both plus and minus).

Authors: Done.

P8L17: rephrase "less peaky"

Authors: The standard deviation of DT2018 EKE is less important than for DT2014 EKE: EKE variations are less important. This section has been improved and details have been added.

P8L22 : could you explain why only th period 2003-2004 can be considered for this assessment ?

Authors: We choose the 2003-2004 period because it is a period over which we have 4 altimeter missions available: TP, J1, EN and GFO. This allow us to keep 2 missions independent for the validation. The remaining 2-altimeter constellation used for the mapping can be compared to the altimeter constellation available before 2003 or for the C3S production. To test the relevance and robustness of the diagnosis, we varied the independent missions over the 2003-2004 period, using alternately J1, EN and GFO as independent missions. The conclusions remain the same. Moreover, it

is a period that has already been studied in Pujol et al, 2016, so we thought it would be interesting to continue over this "reference" period. We also did the study on another more recent year (2017) and the conclusions are similar.

P8L23: The author avoided the nomenclature "two-sat"/"all-sat" up to this point. Can it be also avoided here ? (I think it is the only place where it is used).

Authors: Done

P9L8 : ? is it "COvariance and RMS" ?

Authors: The Taylor skill score (Taylor, 2001) is defined as: $S = \dfrac{4(1+R)}{\left(\dfrac{\sigma_{mod}}{\sigma_{obs}}+\dfrac{\sigma_{obs}}{\sigma_{mod}}\right)^2+(1+R_0)}$

Where $R_0$ is the maximum correlation attainable (hereafter $R_0 = 1$), R is the correlation coefficient between the model and the observations, $\sigma_{mod}$ and $\sigma_{obs}$ are respectively the model and the observations standard deviations.

So it is more correlation and standard deviation than variance and rms.

P9L10 : "altimeter maps" -> "geostrophic current maps"

Authors: Done

P9L12 : lowercase "Variance"

Authors: Done

P9L20 "points" -> "data points"

Authors: Done

P9L20/22 : rephrase "We gain all points".

Authors: Done

P9L26 "in the" repeated

Authors: Done

P10L4 : Why "maximum" correlation ? Does that refer to a selection amongst the neighboring pixels ?

Authors: The processing is detailed in Valladeau et al.,2012. The method is based on a criterion of maximal correlation between tide gauge time series and altimeter gridded products, where the most consistent state of the ocean between both data time series is considered within 300km around tide gauge. The main advantage of this method is to reduce the effect of oceanic variability and the error on the MSS with respect to the same altimeter point.

p10L26 : "a measurementS"

Authors: Done

P11L3, remove "." after "yr" (2x).

Authors: Done

P11L18 "For" -> "for"

Authors: Done

P11L26:28 Why is there no TG validation for the BlackSea ? Explain.

Authors: It has been added.

P12L14 "large" -> "largeR"

Authors: Done

p12l22 "lager" -> "larger"

Authors: Done

P13L8 "for" -> "from"

Authors: Done

P13L26 Biblio ref for eddy tracking, instead of html ?

Authors: The authors have added a reference to a poster presentation which was presented during OSTST 2018 : A Delepoulle et al. and the user manual that describes Mesoscale Eddy Trajectory Atlas product based on DT2018 altimetry products.

* Figures & Tables :
* * *
* Are appropriated and all useful in general. * Small to very small coordinates, axes and colorbar title. Please ensure readability.

Fig 1: What determines the end of the bars for the future ? scheduled lifetime ? please precise.

Authors: Nominal mission life time for missions before launch. Extended lifetime for launched missions. And end of next year for old missions (to account for possible obviated anomalies). Generally derived from CEOS (Committee on Earth Observation Satellites) timeline, or official announcements. Note that the launch dates and lifetimes are constantly in flux, so this figure periodically updated as an indicative timeline either than exact plan from Space Agencies.

Fig 2: Probably the less useful figure. If considered essential, should the figgure be reprocessed with larger bins ? It does not provides many information as for now, except : "more data in the 20km coastal band", "lot of noise in the center" and " a strange, uncommented blue track in the center of East Med". Unless justified otherwise, i suggest to remove this figure.

Authors: The authors have decided to remove this figure.

Fig 3,: caption : rephrase "Loss ones".

Authors: The authors have rephrased this sentence.

Fig 6. Second half of the caption ("Difference of the variance ... ". Does not correspond to the figure (eg. refers to negative values). -> ? missing panel ?

Authors: The missing panel has been added to the figure.

Fig 9: Caption mentions histograms that are not visible on the figure.

Authors: This caption refers to an old version of the figure. It has been corrected.

Fig 10 : use divergent colormap for the panel f,g,h (eg. blue-white-red)

Authors: The authors have changed this figure.

* References :
* * *
* There are many references to work 'in prep.', including on to "In prep. to be submitted to OD in 2016" (Lyard et al.) . Please check with editorial office on the policy as regards reference to unpublished works.

Authors:  The authors have contacted the editor. Here is the answer: In general, please note that "submitted to", "in preparation", "in review", … can be left as is. During typesetting of your manuscript our Typesetters will check all references related to Copernicus Publications for an update. If an update is available our Typesetters will insert it and inform you accordingly.

* The reference style is not homogeneous, with years being given some times at the end, some times after the authors. Please homogenize.

Authors: Done

* There are (many) reference works not provided in the bibliography (eg. Valladeau et al, 201 ; Le Traon et al, 1998, Ducet et al 2000, Le Traon & Ogor 1998 ; Le Traon et al, 2003 ; Lumpkin et al. 2013 ; Taylor, 2001 ; Watson et al, 2015 ; Beckley et al , 2017 ; Dieng et al 2017; Ballarota, in prep ; d'Ovidio 2015.)

Authors: The authors added the missing references.

* Similarly there are (many) references in the biblio that are not mentionned in the text. I do not think it is my duty to revise this for you extensively. Please check carefully.

Authors: The authors have checked.  Many references in the biblio are not mentioned directly in the text but are mentioned in the table 1. The authors did not remove any references.